# Three-dimensional atomic scale electron density reconstruction of octahedral tilt epitaxy in functional perovskites

Yakun Yuan[1,2], Yanfu Lu[1,2], Greg Stone[1,2], Ke Wang[2], Charles M. Brooks[3], Darrell G. Schlom[3,4], Susan B. Sinnott[1,2], Hua Zhou[5] & Venkatraman Gopalan[1,2,6]

Octahedral tilts are the most ubiquitous distortions in perovskite-related structures that can dramatically influence ferroelectric, magnetic, and electronic properties; yet the paradigm of tilt epitaxy in thin films is barely explored. Non-destructively characterizing such epitaxy in three-dimensions for low symmetry complex tilt systems composed of light anions is a formidable challenge. Here we demonstrate that the interfacial tilt epitaxy can transform ultrathin calcium titanate, a non-polar earth-abundant mineral, into high-temperature polar oxides that last above 900 K. The comprehensive picture of octahedral tilts and polar distortions is revealed by reconstructing the three-dimensional electron density maps across film-substrate interfaces with atomic resolution using coherent Bragg rod analysis. The results are complemented with aberration-corrected transmission electron microscopy, film superstructure reflections, and are in excellent agreement with density functional theory. The study could serve as a broader template for non-destructive, three-dimensional atomic resolution probing of complex low symmetry functional interfaces.

---

[1] Department of Materials Science and Engineering, Pennsylvania State University, University Park, PA 16802, USA. [2] Materials Research Institute, Pennsylvania State University, University Park, PA 16802, USA. [3] Department of Materials Science and Engineering, Cornell University, Ithaca, NY 14853, USA. [4] Kavli Institute at Cornell for Nanoscale Science, Ithaca, NY 14853, USA. [5] Advanced Photon Source, Argonne National Laboratory, Lemont, IL 60439, USA. [6] Department of Physics, Pennsylvania State University, University Park, PA 16802, USA. Correspondence and requests for materials should be addressed to H.Z. (email: hzhou@aps.anl.gov) or to V.G. (email: vxg8@psu.edu)

Complex oxides interfaces have become a vibrant research focus in condensed matter physics and material science[1–5], since they are a fertile playground for emergent phenomena such as, magnetism[6], ferroelectricity[7], interface charge transfer[8], two-dimensional free electron gases[9], superconductivity[10], and topological states[11] through strategies in modern materials design, including strain tuning[12–14], artificial layering[15,16], spatial confinement[17], and interfacial coupling[18–25]. Control of octahedra tilts in complex oxides via film-substrate interface design, or tilt epitaxy, has been predicted to be a powerful knob for tuning various functional properties, including inversion symmetry breaking[26–28], magnetism[18,22,29], and electronic orders[30]. Although the tilt epitaxy promises a potentially wonderful route for designing these functionalities in various materials, the experimental reports on realizing tilt epitaxy are still very limited. So far, the tilt epitaxy has been used to stabilize polar distortions in metallic ultrathin nickelates films[20] and to manipulate magnetic anisotropy in $SrRuO_3$[18,29] and $La_{2/3}Sr_{1/3}MnO_3$[22]. However, in these works, only in-phase octahedra tilt along one of the three crystallographic axes are experimentally resolvable. Moreover, strain and substrate termination effects can convolute with tilt epitaxy, which remain unexplored.

In general, there are three outstanding challenges in implementing tilt epitaxy. The first is that substrate tilts can transfer into the film only to within ~10 unit cells, thus necessitating ultrathin films to observe these dramatic effects. Secondly, experimentally determining the complete three-dimensional (3D) structure of such tilt epitaxy interfaces with atomic resolution is quite a formidable task. Direct aberration-corrected scanning transmission electron microscopy (STEM) is now routinely used for probing atomic structures with picometers metrology; however, they probe the potential of two-dimensional projections of atomic columns, and deconvolving the information along depth direction is a challenge[22], as we illustrate in this work. Coherent Bragg rods analysis (COBRA)[31–41], which reconstructs 3D electron density with atomic resolution based on a phase retrieval algorithm taking advantage of the interference between the diffracted X-ray beams from the thin film and the substrate, is a promising technique for such purpose. COBRA requires no special sample preparation (such as in STEM) and is readily applicable to any epitaxial system with film thickness <20 nm. However, previous COBRA studies have mostly focused on systems with high symmetry, e.g., $4mm$ point group, and heavy cations. The complete 3D analysis of oxygen octahedra for a low symmetry system is still an outstanding challenge. Other emerging 3D imaging techniques (see Supplementary Table 1) include coherent diffraction imaging[42], tomography[43], topography[44], ankylography[45] using X-ray or electrons, each with its own merits and drawbacks. The third challenge is to be able to deconvolve the influence of the tilt epitaxy from that of strain and surface termination effects that may coexist.

In this work, we tackle all three of these outstanding challenges. We study ultrathin films of a prototypical perovskite with a complex tilt pattern, namely calcium titanate on various substrates that provide a range of tilt and strain states. We report the atomic scale 3D reconstruction of the electron density across these low symmetry epitaxial complex oxides interfaces by COBRA, the first such feat where both substrate and film possess three octahedral tilts in addition to polar distortions. The reconstruction requires high quality mapping of diffractions in a large reciprocal space volume and generalized computer routines for handling the large experimental data set. Specifically, we present COBRA reconstructed electron densities (EDs) of ultrathin epitaxial $CaTiO_3$ films on $NdGaO_3(110)$, $DyScO_3(110)$ and $La_{0.29}Sr_{0.71}Al_{0.65}Ta_{0.35}O_3(001)$ (LSAT) substrates, each offering a unique combination of strain and octahedral tilt patterns across

the interface. Combining COBRA studies with complementary scanning transmission electron microscopy (STEM) and density functional theory (DFT) reveals the distinct roles of tilt epitaxy, strain and surface termination. We find that, in addition to epitaxial strain effect inducing polar distortion in the film, the tilt epitaxy monoclinically distorts the film and clamps the in-plane oxygen octahedral tilts of $CaTiO_3$ on LSAT substrate, giving rise to significantly higher polar transition temperatures (>900 K) in ultrathin $CaTiO_3$ films (8 u.c. or ~3.0 nm thick) than previously reported for thicker films (>10 nm)[46–49]. Moreover, an unexpected out-of-plane polarization is observed in tensile strained $CaTiO_3$ thin films with directions dictated by the interfacial valence mismatch. These tilt epitaxy as well as valence mismatch effects should be present in all epitaxial complex oxides systems and strongly mediate the properties of ultrathin epitaxial films, which provide new routes to artificially control materials functionalities.

## Results

**Interplay of strain and octahedral tilts at the interface.** $CaTiO_3$ has a centrosymmetric $Pnma$ space group and is comprised of corner-shared oxygen octahedral network with interstices filled by calcium and titanium atoms. In bulk form, the oxygen octahedra exhibit out-of-phase oxygen octahedra tilts of 9.1°($a^-$) about the $[100]_{pc}$ and the $[001]_{pc}$ axes (pc: pseudocubic), and an in-phase ($b^+$) tilt of 9.2° about the $[010]_{pc}$ axis, denoted as $a^-b^+a^-$ using the Glazer notation[50,51]. Previous literatures using STEM imaging and dielectric measurements on 10 nm or thicker biaxially strained epitaxial $CaTiO_3$ films on different substrates have shown that a paraelectric to ferroelectric phase transition occurs with a tensile strain of >1.1%, leading to an in-plane polarization[46–49]. However, we find that the story changes dramatically for ultrathin (8 u.c. or ~3.0 nm) films used in this study, where out-of-plane polarization also arises, Curie temperatures are significantly higher, the effect of chemical termination at the interface and geometric oxygen octahedral tilt mismatch between substrate and $CaTiO_3$ becomes important. To ascertain the above effects, an 8 u.c. of $CaTiO_3$ was epitaxially grown on $NdGaO_3(110)_{or}$ (or: orthorhombic), $DyScO_3(110)_{or}$ and $LSAT(001)_{pc}$ substrates, with a tensile strain of 1.1%, 3.3% and 1.2%, respectively, by using Molecular Beam Epitaxy (see Methods). Similar to $CaTiO_3$, $NdGaO_3$ and $DyScO_3$ possess $a^-b^+a^-$ tilt pattern, with out-of-phase tilts ($a^-$) of 10.3° and 15.0°, respectively, and in-phase tilts ($b^+$) of 9.8° and 13.0°, respectively. LSAT adopts a simple cubic structure with 0° tilts (or $a^0a^0a^0$ under Glazer notation). The abrupt tilt mismatch in octahedral tilt angles across the interfaces are schematically displayed in Fig. 1a, b, where $NdGaO_3$ and $DyScO_3$ prefer enhanced angles along $[100]_{pc}$ and $[010]_{pc}$ axes (Fig. 1a), while LSAT tends to suppress those angles (Fig. 1b).

The 3D structures of the above systems were investigated using the COBRA method by modeling the interference between diffractions from ultrathin epitaxial $CaTiO_3$ films and the three substrates. As shown in Fig. 1c, the synchrotron X-ray diffraction from such epitaxial system form crystal truncation rods (CTRs) at integer $H$, $K$ values (in-plane directions of the film) with a continuous distribution along the $L$ (thickness direction of the film) in the reciprocal space. By rotating the sample about its surface normal ($L$) axis, the CTRs intersect with the Ewald sphere at different $L$ positions, and the diffraction in the full reciprocal space can be mapped out. Measurements on $CaTiO_3/NdGaO_3$, $CaTiO_3/DyScO_3$ and $CaTiO_3/LSAT$ were performed at both room temperature and 30 K (at which all films are in polar state). A phase-retrieval algorithm (COBRA) is then employed to reconstruct the 3D electron density in real space[31–41] (See Methods and Supplementary Note 1 for experimental details).

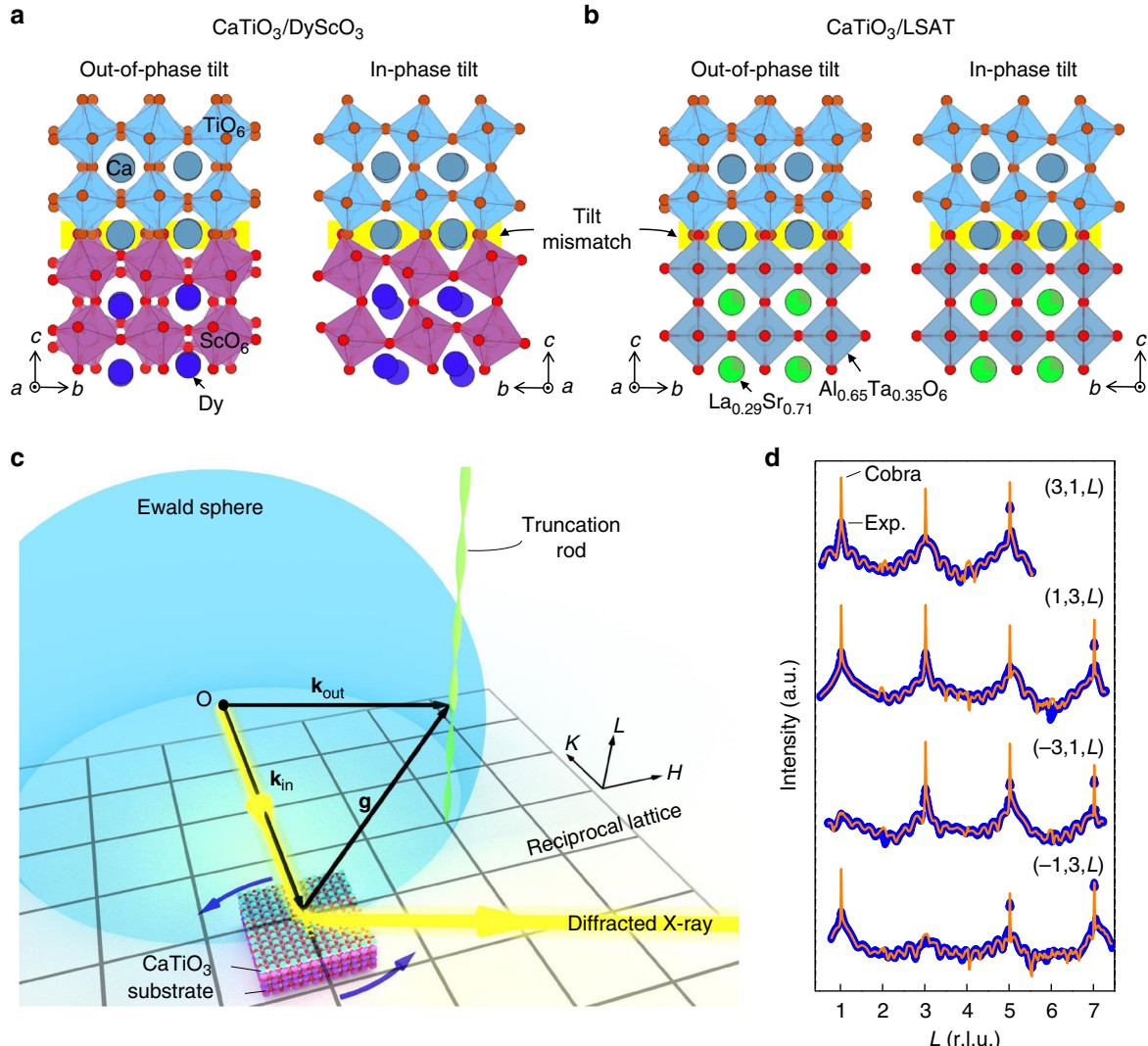

**Fig. 1** The schematics of film systems and CTRs measurement. Schematic of tilt mismatch across $CaTiO_3$ and substrate interfaces for **a** $CaTiO_3$/$DyScO_3$ **b** $CaTiO_3$/LSAT. The yellow highlighted area represents the abrupt mismatch in in-plane oxygen octahedra tilt angles. **c** Experimental geometry of synchrotron X-ray diffraction used in the coherent Bragg rods analysis (COBRA) is schematically shown. The diffraction intensities along crystal truncation rods (CTRs) are mapped out by varying their intersect positions with Ewald sphere through rotating the sample about its surface normal. ($H$, $K$, $L$) is the coordinates in reciprocal lattice. **d** Several examples of measured crystal truncation rods, labeled as ($H$, $K$, $L$), where the intensity versus the reciprocal vector $L$ is shown (blue dots) for $CaTiO_3$/$NdGaO_3$ at room temperature, as well as the corresponding COBRA fits (orange solid lines). The reciprocal units are under $2 \times 2 \times 2$ pseudocubic notation. Note that for COBRA, the substrate Bragg peaks intensities (the trees) are not as important as the diffraction structure between the peaks (the forest floor)

Figure 1d displays typical COBRA fits (orange solid lines) to experimental CTRs (blue dots) measured on $CaTiO_3$/$NdGaO_3$ at room temperature.

**3D atomic structure by COBRA and DFT**. The reconstructed 3D electron density maps at 30 K for $CaTiO_3$/$NdGaO_3$, $CaTiO_3$/$DyScO_3$ and $CaTiO_3$/LSAT are shown respectively in Fig. 2a–c. All the atomic positions, including oxygen atoms, are clearly visible as discrete peaks in the electron density maps. The high quality of the electron densities suggests the films are epitaxial and of high crystallinity. We first focus on the domain states of the three systems. Since both $NdGaO_3$ and $DyScO_3$ have the same space group ($Pnma$) as $CaTiO_3$, the epitaxially grown 8 u.c. $CaTiO_3$ is expected to follow the crystallographic orientation of substrates to minimize the interfacial energy. Indeed, a mono-domain of 8 u.c. $CaTiO_3$ on $NdGaO_3$ and $DyScO_3$ is

confirmed by the symmetry exhibited in CTRs (See Supplementary Note 2), as well as the consistent oxygen octahedral tilt pattern ($a^-b^+c^-$) across the interfaces, as seen in Fig. 2a, b. However, since LSAT possesses an effectively higher (cubic) symmetry ($Fm\overline{3}m$) than $CaTiO_3$, four symmetry equivalent domains exist within the X-ray probe area (~500 µm) with equivalent fractions, as evidenced by the symmetry of the measured CTRs (See Supplementary Note 2). Therefore, the reconstructed electron densities of $CaTiO_3$/LSAT contains folded structural information, as shown in Fig. 2c, which is the result of spatially translating $CaTiO_3$ into a single pseudocubic unit cell[52]. A mixed tilt pattern of $a^-b^+c^-$/$a^+b^-c^-$ is observed for $CaTiO_3$ on LSAT. (See Supplementary Note 3 for details on structural folding and tilt pattern) The structural details of the three systems can be better visualized by breaking down the 3D electron densities into different slices of atomic planes. The $ac$, $bc$, and $ab$ slices through the $TiO_2$ atomic planes of $CaTiO_3$/$NdGaO_3$ are

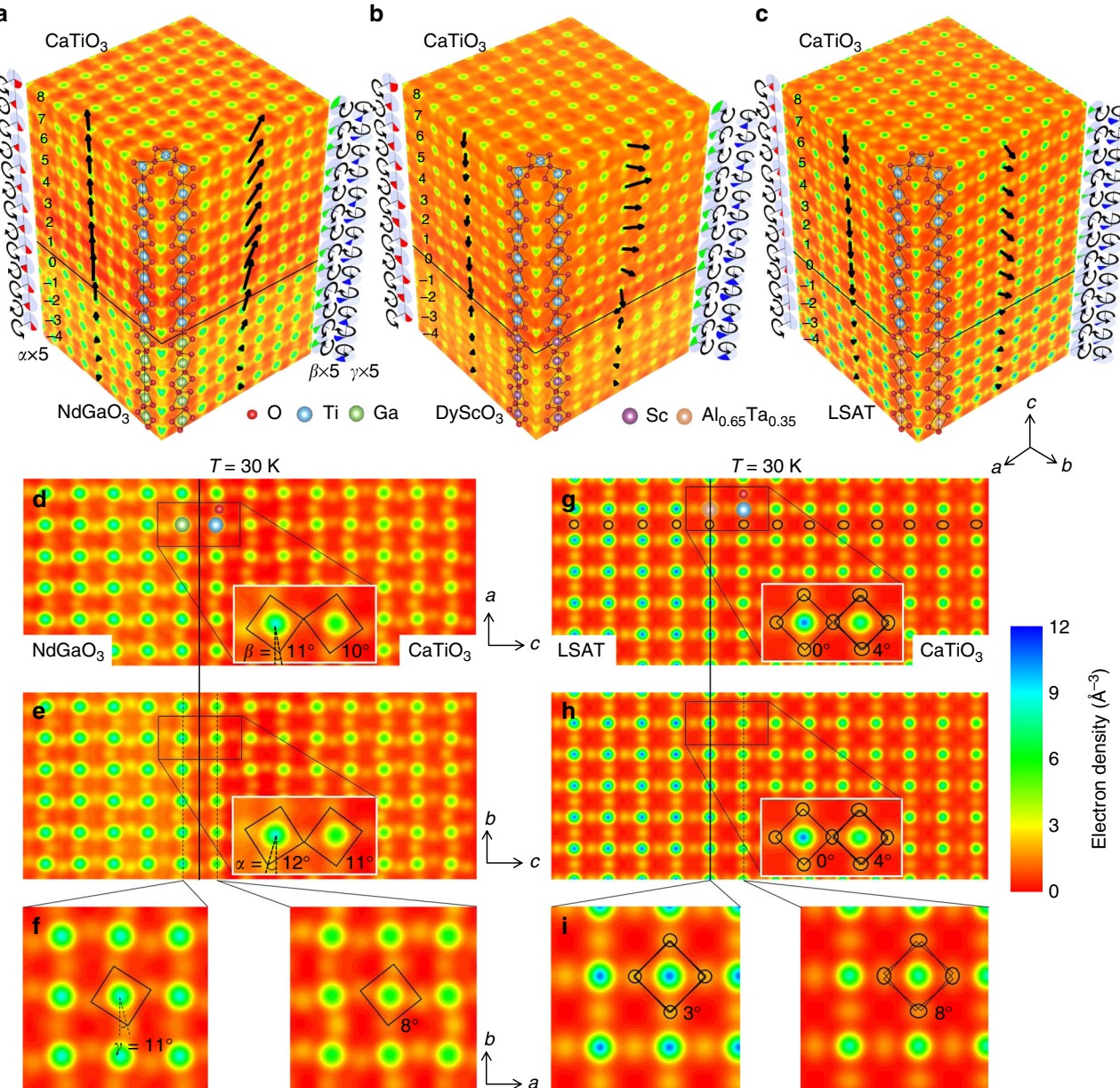

**Fig. 2** 3D electron densities reconstructed by COBRA at 30 K. Three-dimensional electron densities for **a** CaTiO$_3$/NdGaO$_3$, **b** CaTiO$_3$/DyScO$_3$, **c** CaTIO$_3$/LSAT at 30 K reconstructed by coherent Bragg rods analysis (COBRA). Polarizations projections on $ac$ and $bc$ planes are displayed on corresponding faces. Oxygen octahedra tilts evolution, $\alpha$, $\beta$, and $\gamma$, are magnified by 5 times and displayed by red, green, and blue pie charts along each edge of electron densities. One of the four equivalent CaTiO$_3$ domains on LSAT are plotted for convenience of comparison. Two-dimensional slices of **d** $ac$, **e** $bc$, and **f** $ab$ planes for CaTiO$_3$/NdGaO$_3$ and the corresponding plots **g**, **h**, **i**, for CaTiO$_3$/LSAT are shown, respectively. The interconnected oxygen networks are sketched by black squares, which tends to clamp the in-plane tilts ($\alpha$ and $\beta$) in CaTiO$_3$ through interface. While, the $\gamma$ tilt in CaTiO$_3$ is unaffected

respectively displayed in Fig. 2d–f. The interconnected oxygen octahedra networks can be clearly seen in all three slices, as sketched by black squares. The tilt angles of the oxygen octahedra, $\alpha$, $\beta$, and $\gamma$, are indicated on each slice. Here we focus on the impact of tilt mismatch on those angles across the interface. The $\beta$ angle of the rightmost GaO$_2$ layer in $ac$ slice is about $11 \pm 1°$, while its neighboring TiO$_2$ layer next to the interface developed a tilt of $10 \pm 1°$, which is close to the value of NdGaO$_3$ substrate and slightly larger than that of bulk CaTiO$_3$. Similar behavior is also observed in $bc$ slice, where $\alpha$ angles of $12 \pm 1°$ and $11 \pm 1°$ are measured for neighboring GaO$_2$ and TiO$_2$ layers across the interface. This suggests that the interconnected octahedra network in $ac$ and $bc$ slices can effectively propagate the in-plane tilts ($\alpha$ and $\beta$) of substrate into the epitaxial film. However, with this

picture of tilt epitaxy, one would expect the out-of-plane $\gamma$ tilt to be relatively decoupled across the interface. Indeed, as shown in Fig. 2f, the GaO$_2$ and TiO$_2$ layers near the interface give $\gamma$ angles of $11 \pm 1°$ and $8 \pm 1°$, showing a relatively large change. CaTiO$_3$ on DyScO$_3$ possesses very similar structural characteristics as on NdGaO$_3$, where the exact same behavior can be observed; thus it is not shown here. The slices of folded 3D electron density for CaTiO$_3$/LSAT are shown in Fig. 2g–i. As discussed in Supplementary Note 3, the tilts of the oxygen octahedra will give rise to broadened oxygen peaks, representing multiple equivalent oxygen atom positions at corresponding oxygen sites. The broadening of oxygen peaks is indicated by the solid ellipses, which mark the contours of oxygen peaks. In this case, the tilt angles can be extracted by fitting the splitting of the oxygen atoms at each site.

The $ac$ slice in Fig. 2g shows an increase of ellipticity of oxygen peaks from the interface to the surface of the film, indicating an increase of tilt angles. The $bc$ slice (Fig. 2h) shows the same behavior due to the symmetry equivalency of $a$ and $b$ axes of the folded electron density. The in-plane tilt angles ($\alpha$ and $\beta$) for the neighboring $Al_{0.65}Ta_{0.35}O_2$ and $TiO_2$ layers are respectively $0 \pm 2°$ and $4 \pm 2°$, yielding significantly smaller in-plane tilt magnitudes in $CaTiO_3$ film compared to its bulk values. This again agrees with the tilt epitaxy effect through the interface. Similarly, the $\gamma$ angles are $4 \pm 3°$ for $Al_{0.65}Ta_{0.35}O_2$ and $8 \pm 3°$ for $TiO_2$ layers near the interface, confirming that the $\gamma$ tilt of the $CaTiO_3$ film is decoupled from the substrate.

Quantitative analysis of the 3D electron densities is performed as follows. By fitting each peak in the electron density with 3D Gaussian functions, the 3D coordinates of atoms in each unit cell from 5 u.c. beneath the substrate to the surface of the 8 u.c. thick film are extracted. Electrical polarization vectors are calculated by using cations displacements relative to anions (oxygen) and their nominal charges. The projections of polarization vectors on $ac$ and $bc$ planes are plotted as black arrows on the corresponding faces of the electron density maps (Fig. 2a–c), depicting a polar phase at 30 K in films on all three substrates. For convenience of comparison, one of the four equivalent domains on LSAT is plotted.

The polarizations vector evolution along the growth direction is discussed next. As illustrated in Fig. 3a–c, $CaTiO_3$ films on $DyScO_3$, $NdGaO_3$, and LSAT, respectively exhibit average in-plane polarizations of $-20.6 \pm 2.1$, $-14.9 \pm 1.8$, and $13.5 \pm 2.5 \mu Ccm^{-2}$ along the $a$-axis (green circles). There is no measurable polarization along the $b$-axis (blue circles). The magnitudes of the in-plane polarizations qualitatively agree with the larger tensile strain states on $DyScO_3$ (3.3%) and smaller tensile strain on $NdGaO_3$ (1.1%) and LSAT (1.2%); the numbers however deviate from previous theory predictions on bulk state[47], which will be addressed in detail further on. Most interestingly, unexpected out-of-plane polarization components are observed in $CaTiO_3$ on all three substrates (red circles in Fig. 3a–c).

Remarkably, COBRA indicates that the directions of the out-of-plane polarizations appear to be dictated by the substrate terminations, due to a valence mismatch effect[21]. The NdO layer termination of $NdGaO_3$ substrate, as indicated by the black line in Fig. 2a, with a valence mismatch of $+1$, prefers an out-of-plane polarization towards the $+c$ direction. In contrast, electron density in Fig. 2b, c (black lines) indicate that an $ScO_2$ and $Al_{0.65}Ta_{0.35}O_2$ ($BO_2$) termination with a valence mismatch of $-1$ and $-0.3$, respectively, result in an out-of-plane polarization direction of $-c$. These COBRA reconstructed surface terminations are confirmed experimentally using transmission electron microscopy as shown in Supplementary Figure 5.

The combination of in-plane and out-of-plane polarization components determine the polarization vectors as shown, which lie in a single mirror plane, indicating the monoclinic symmetry with $m$ in the $ac$ plane. Note that to reveal this low symmetry distortion by COBRA, one has to collect enough truncation rods; in this particular case, up to 47 CTRs for each system were collected. Further, in contrast to previous COBRA algorithm, the current phase retrieval algorithm was generalized to include all possible crystallographic symmetries.

Quantitative structural analysis of the oxygen octahedral tilt angles $\alpha$, $\beta$, and $\gamma$, respectively about the $a$, $b$ and $c$ axes, are performed by analyzing oxygen atom peaks in each $BO_6$ octahedron and are depicted as pie charts along the edges of the electron density maps (Fig. 2a–c) as well as plotted in Fig. 3d–f as green ($\alpha$), blue ($\beta$), and red ($\gamma$) circles. COBRA reveals a gradual change in the $\alpha$ and $\beta$ values and a relatively drastic change in the $\gamma$ values across the interfaces for all three

systems, as suggested by the 2D slices in Fig. 2d–i. This is expected due to the tilt epitaxy effect on the $\alpha$ and $\beta$ angles between the substrates and their corresponding $CaTiO_3$ films through the shared interfacial oxygen atoms (as illustrated in Fig. 1a, b), while the $\gamma$ angles are not affected, allowing for a drastic change across the interface.

The above reconstructed 3D oxygen tilts and polar displacements in the film suggest the role of substrate strain, substrate oxygen octahedral tilts, and substrate terminations. To understand and deconvolve these effects, density functional theory (DFT) calculations were performed (see Methods). The DFT results for the three components of the polarization and the octahedral tilts for each film system are plotted using solid lines in Fig. 3a–f, showing excellent agreement between the calculations and COBRA experiments. To uncouple the changes in the octahedral tilts induced by tilt epitaxy from the changes due to a pure biaxial strain, bulk calculations on strained $CaTiO_3$ (with no interfaces) were performed. The differences in the magnitude of in-plane octahedral tilts, $|\Delta\alpha| + |\Delta\beta|$, between the film and the substrate were smaller in the case where the tilt epitaxy effect is present versus when the substrate simply imparts a biaxial strain. (See Supplementary Table 2 for summary of tilt angles) For example, for the $CaTiO_3/NdGaO_3$ film system, $|\Delta\alpha| + |\Delta\beta| = 0.63°$ from DFT which agrees well with the measured $0.7 \pm 0.5°$ from the COBRA reconstruction. In contrast, it is $1.21°$ from the strained bulk calculation where there is no interface tilt epitaxy effect, thus indicating the important role of tilt epitaxy in minimizing the in-plane tilts difference ($|\Delta\alpha| + |\Delta\beta|$) across the interface. A similar trend for $|\Delta\alpha| + |\Delta\beta|$ is seen for the other two film systems: $6.33°$ (DFT including tilt epitaxy) and $3.7 \pm 1.3°$ (COBRA) versus $8.5°$ (DFT bulk without tilt epitaxy) for the $CaTiO_3/DyScO_3$; and respectively, $8.58°$ and $12.7 \pm 1.0°$ versus $18.88°$ for the $CaTiO_3/LSAT$ system. These qualitatively excellent and quantitatively good comparisons between DFT and COBRA confirm the tilt epitaxy and valence mismatch effects on the in-plane and out-of-plane polarization components of epitaxial $CaTiO_3$ thin films.

**Interface controlled polarization state.** The influence of interfacial tilt epitaxy and valence mismatch effects is expected to diminish as the epitaxial film thickness increases and should be much more prominent in ultrathin films. Fig. 3g, h show the polarization comparison between literature values on thick $CaTiO_3$ films (>10 nm)[46–49] and values on ultrathin films studied in this work, where theoretical phase field simulations (green lines), DFT (yellow lines) on strained bulk $CaTiO_3$, dielectric measurement results on >10 nm thick $CaTiO_3$ films (green squares, blue triangles), as well as COBRA (open stars) and DFT (closed stars) results on ultrathin (8 and 6 u.c., respectively) epitaxial $CaTiO_3$ films are presented. The comparison of in-plane polarization (Fig. 3g) shows a perfect match between experimentally measured polarization on thick $CaTiO_3$ films (green squares and blue triangles) and theoretical calculation (green and yellow lines) on strained bulk $CaTiO_3$, where no interface is present. DFT calculations (closed stars) on ultrathin epitaxial $CaTiO_3$ films agree well with COBRA results (open stars). The slight increase in the in-plane polarization at 3.3% strain for ultrathin films is consistent with the fact that higher tensile strain favors larger in-plane polarization. The ultrathin films give significantly higher in-plane polarization under 1.1–1.2% strain ($NdGaO_3$ and LSAT) and lower polarization under 3.3% strain ($DyScO_3$) as compared to thick films, which suggests that the interfacial tilt epitaxy effect can dominate the properties of these films. Similarly, in Fig. 3h, tensile strained bulk $CaTiO_3$ exhibits zero out-of-plane polarization components, while ultrathin films show a clear non-zero

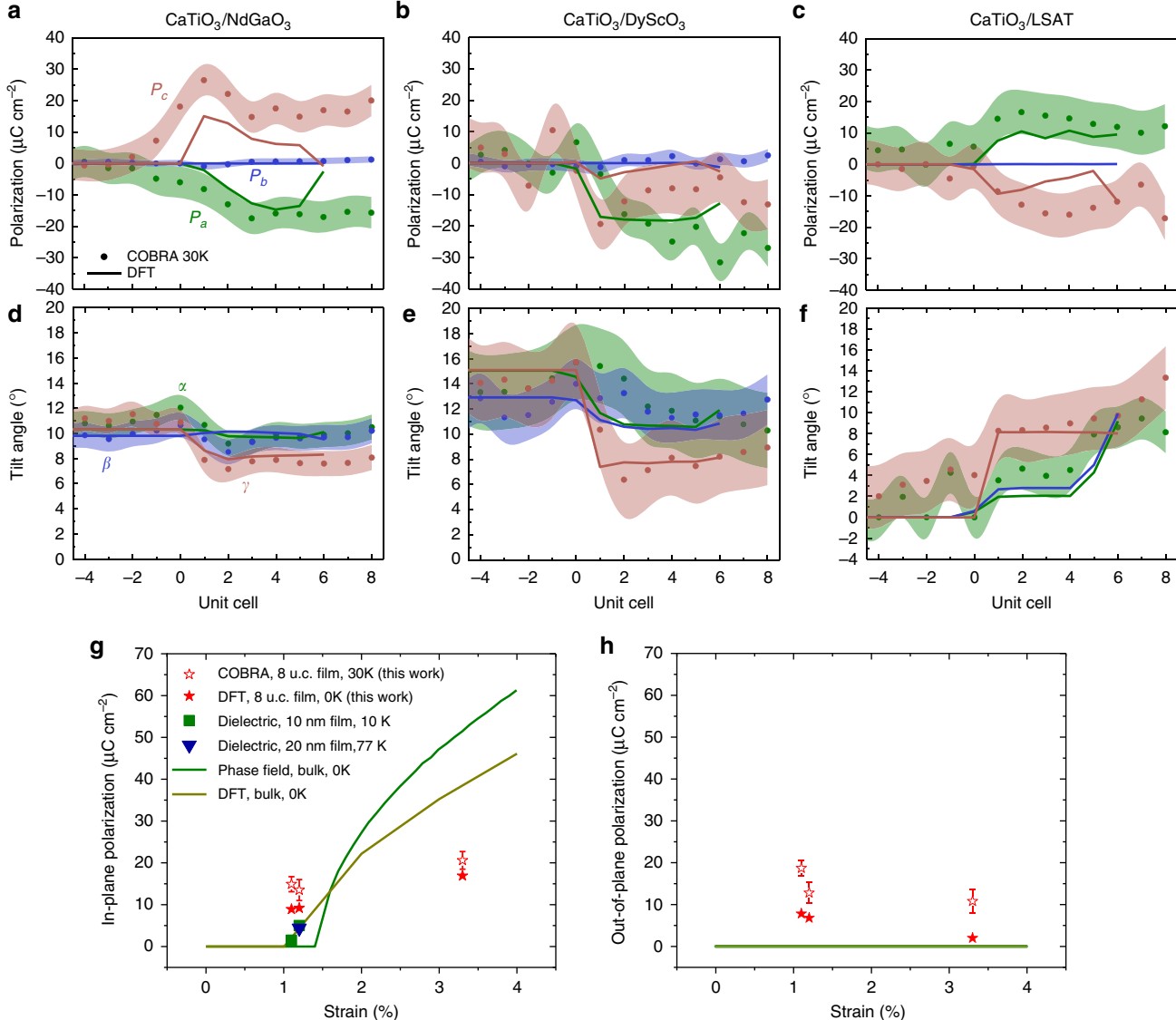

**Fig. 3** Tilt angles and polarizations in ultrathin CaTiO₃ films. **a-c** Polarization components, $P_a$ (green), $P_b$ (blue), and $P_c$ (red), as well as **d-f** quantified tilt angles, $\alpha$ (green), $\beta$ (blue), and $\gamma$ (red), extracted from coherent Bragg rods analysis (COBRA) (dots) and density functional theory (DFT) (solid lines) are compared for the films on NdGaO₃, DyScO₃, and LSAT. The experimental errors are estimated by comparing COBRA results for substrates to their bulk reference values and are indicated by the shaded area surrounding the COBRA data dots. **g** The polarization state of ultrathin CaTiO₃ are compared to literature reports on thick samples (>10 nm). Theoretical works by phase field simulation (green line[47]) and DFT (yellow line[46]) were performed on bulk CaTiO₃ at 0 K. Experimental works on epitaxial thin film were measured by low temperature dielectric measurements. Green squares show measurements on 10 nm CaTiO₃ at 10 K[49]. Blue triangle shows measurement on 20 nm CaTiO₃ at 77 K[48]. The in-plane polarizations obtained by COBRA method at 30 K for 8 u.c. films and DFT for 6 u.c. films are respectively plotted using red open and closed stars. **h** Phase field (green line) and DFT (yellow line) show no out-of-plane polarization for strained bulk CaTiO₃. COBRA (red open star) and DFT (red closed star) reveal a clear non-zero out-of-plane polarization in ultrathin CaTiO₃ films

polarization with decreased magnitude at 3.3% tensile strain. This non-zero polarization in ultrathin films again display the effect of interfacial tilt epitaxy and its competition with strain. We also notice that the tilt angles of these ultrathin films change significantly over the first few unit cells, and then tend to relax on approaching the surfaces; however, they do not fully relax to the bulk value within the 8 u.c. This explains the reduced out-of-plane polarization on approaching the film surface. (The outermost u.c. has surface effects and is not included in the discussion here.) With the tilt epitaxy being the driving force, the long-range electrostatic interaction also plays a role in stabilizing this out-of-plane polarization state. The larger out-of-plane polarization in the first few layers favors the polarization with the same direction

in the rest of the film, while a depolarization field leads to the relaxation of the out-of-plane polarization from the interface to the film surface.

**Probing tilt epitaxy by STEM**. Aberration-corrected scanning transmission electron microscopy (AC-STEM) was also employed to confirm the structure determined from COBRA reconstruction at room temperature, to the extent possible by STEM. Atomic resolution annular bright field scanning transmission electron microscopy (ABF-STEM) of the above three epitaxial systems along [010]₍pc₎ (b-axis) zone axis reveal high quality CaTiO₃ thin films that are epitaxially grown on three different substrates, as

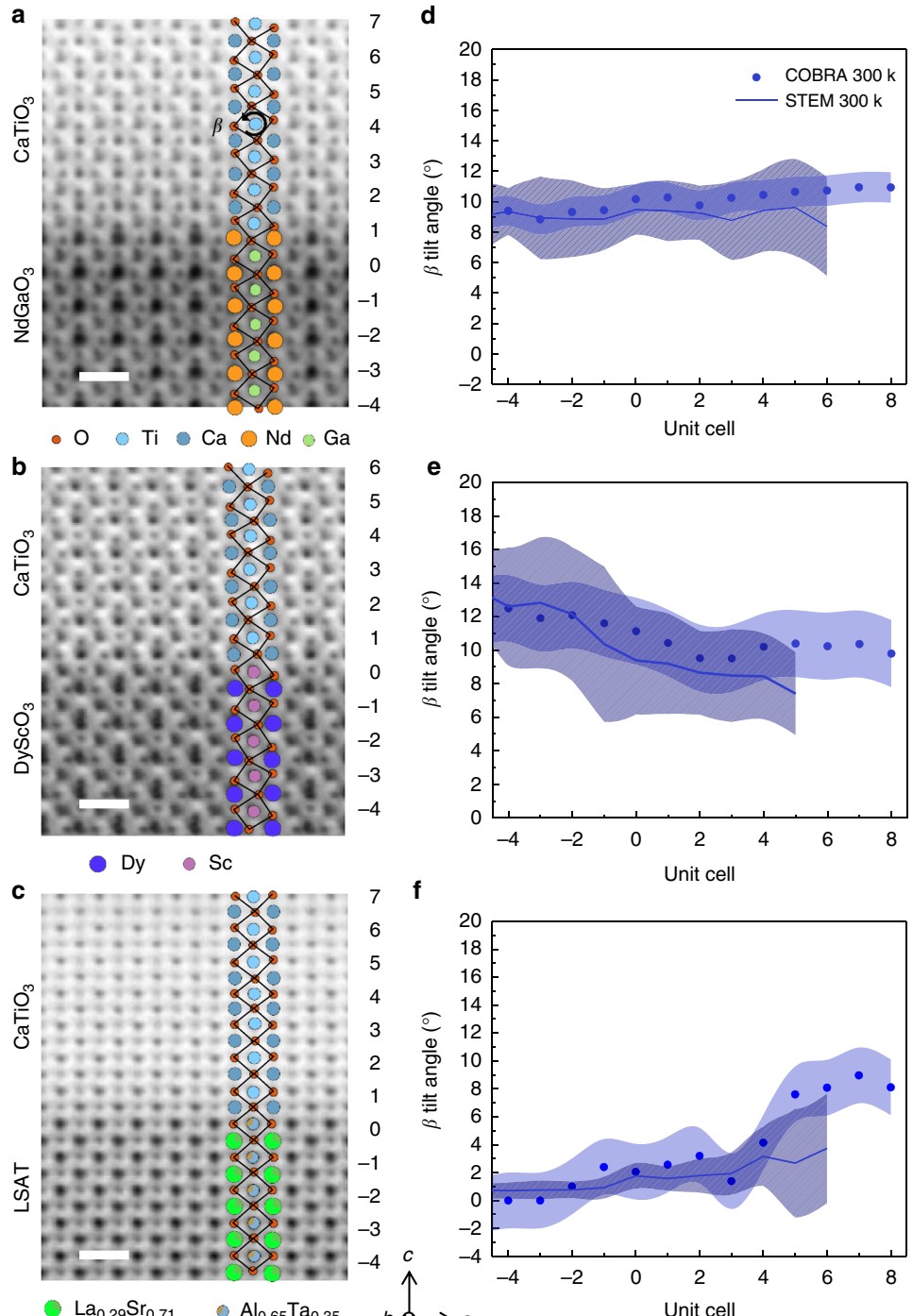

**Fig. 4** Room temperature evolution of $\beta$ angles across interfaces. High resolution annular bright field scanning transmission electron microscopy (ABF-STEM) images with oxygen atoms at room temperature for **a** CaTiO$_3$/NdGaO$_3$, **b** CaTiO$_3$/DyScO$_3$, **c** CaTiO$_3$/LSAT along [010]$_{pc}$ (*b*-axis) zone axis. Different unit cells are labeled by numbers on the right side of the STEM images. The scale bar is 5 Å. The $\beta$ angles (blue lines) resolved by STEM for different unit cells along *c* direction are plotted in **d**, **e**, and **f**, respectively for above systems. As comparison, room temperature coherent Bragg rods analysis (COBRA) data are plotted using blue dots. Experimental errors for STEM (line shaded area) are taken to be the standard deviation of the $\beta$ values along *a*-axis over around 20 unit cells. The errors for COBRA data (light blue shaded area) are obtained by comparing substrates $\beta$ values to their bulk reference

shown in Fig. 4a–c. The substrate surface termination of NdO (AO) for NdGaO$_3$, ScO$_2$ (BO$_2$) for DyScO$_3$, and Al$_{0.65}$Ta$_{0.35}$O$_2$ (BO$_2$) for LSAT are confirmed by energy dispersive spectroscopy mapping (see Supplementary Figure 5), which are in excellent agreement with the COBRA data. Oxygen atoms in all three

systems are clearly visible and display a consistent $b^+$ tilt pattern. In Fig. 4d–f, the evolution of the $\beta$ values obtained from AC-STEM (blue lines) shows a gradual change across the interfaces and are in good agreement with the analysis of room temperature electron densities reconstructed by COBRA (blue dots). Similar to

the low temperature results, a significant suppression of CaTiO$_3\beta$ values is observed on the LSAT substrate. Supplementary Note 4 shows the complete room temperature COBRA results. Since STEM probes 2D projection of atom columns, constructing 3D information relies on images along multiple zone axes. Supplementary Notes 5 and 6 show STEM analysis on $[100]_{pc}$ and $[110]_{pc}$ zone axes. While the qualitative agreement between STEM and COBRA data is reasonable, the results illustrate clearly the challenge in STEM in quantitatively determining the out-of-phase tilts $\alpha$ and $\gamma$ angles that COBRA is easily able to do.

**Probing the polar state using SHG method**. To confirm the point group symmetry resolved by COBRA, and to investigate the polar transition Curie temperature ($T_C$), we employ optical second harmonic generation (SHG), schematically shown in Fig. 5a, where a linearly polarized light $\lambda = 800$ nm is incident onto the sample at an angle $\theta$, and the second harmonic signal at $\lambda = 400$ nm is measured. In Fig. 5b, temperature dependent SHG signal reveals Curie temperatures $T_C$ of 200 and 350 K, respectively, for 8 u.c. CaTiO$_3$ on NdGaO$_3$ and DyScO$_3$, which are significantly higher than that of 70 and 170 K, reported in thick films (10 nm) in literatures[47,49]. Most strikingly, CaTiO$_3$ on LSAT exhibits significant SHG signal from 4 up to 900 K. However, literature reports a $T_C$ of 140–190 K for thick CaTiO$_3$ films (>10 nm) on LSAT[48,49]. Room temperature electron densities reconstructed by COBRA confirm a paraelectric state for CaTiO$_3$ on NdGaO$_3$, a weak polarization of the film on DyScO$_3$ and a large polarization of the film on LSAT (see Supplementary Figure 6). The large enhancement of polar transition temperature, $T_C$, in the 8 u.c. CaTiO$_3$ films in this work as compared to thicker films is a direct result of the interfacial tilt epitaxy effect, which stabilizes the polar phonon soft mode against its competing centric oxygen octahedral modes through interfacial coupling[53]. For the same reason, 8 u.c. CaTiO$_3$ on LSAT exhibits the highest $T_C$ (>900 K) among all three systems, due to its smallest tilt angles arising from the tilt epitaxy on a substrate with no tilts. Symmetry of CaTiO$_3$ is determined by SHG polarimetry, where s-/p-polarized SHG signal components, $I_{2\omega,s}$ and $I_{2\omega,p}$, are measured as a function of incident polarization angle $\varphi$ (see Methods). Theoretical modeling (see Fig. 5c and Supplementary Fig. 10) indicates a single domain point group of $m$ for the films on NdGaO$_3$ and DyScO$_3$ and four equivalent $m$ domains with each domain fraction of ~0.25 for the film on LSAT. These results are consistent with the polar states extracted from the COBRA electron density maps.

## Discussion

This work demonstrates that tilt epitaxy, namely, slight changes in octahedral tilts in perovskites through interfacial tilt control can dramatically influence the functional properties of ultrathin films. The reconstructed 3D electron density in ultrathin films clearly reveal the intertwining roles of tilt epitaxy, substrate strain, and substrate surface terminations. These give rise to unexpected out-of-plane and in-plane polarization components, as well as large enhancement of polar Curie temperatures $T_C$. An important highlight of this work is the development of the Fourier phase retrieval COBRA method to successfully reconstruct the 3D atomic resolution structure of low symmetry complex oxides interfaces with all independent octahedral tilts and polarization vectors in both the film and substrate, the most complex low symmetry interface structure reconstructed to date by this technique. With the development of high-energy surface X-ray diffraction[54] that enables the rapid capture of large portions of 3D Bragg rods in reciprocal space, COBRA measurements will become much more efficient and widespread in their application to oxide thin films and heterostructures. This work will motivate

progress in the fledgling field of tilt epitaxy engineering in ultrathin perovskite films, and more broadly, provide a powerful non-destructive tool with atomic resolution for probing the electron density of complex functional interfaces.

## Methods

**Sample growth by molecular beam epitaxy**. Epitaxial CaTiO$_3$ thin films were grown on DyScO$_3$, NdGaO$_3$, and LSAT using reactive molecular-beam epitaxy (MBE) in a Veeco GEN10 system equipped with reflection high-energy electron diffraction (RHEED) and utilizing a background partial pressure of ~$5 \times 10^{-7}$ Torr of distilled ozone. Calcium was evaporated from an effusion cell and titanium from a Ti-Ball™ sublimation source. The fluxes of the constituent elements, calcium and titanium, were measured using a quartz crystal monitor (QCM) and typical values for each element were around $2 \times 10^{13}$ atoms cm$^{-2}$s$^{-1}$. The 8 u.c. CaTiO$_3$ films were grown at a temperature of 650 ℃ by co-depositing CaO and TiO$_2$. The starting fluxes of the Ca and Ti molecular beams were initially determined by QCM and then the calcium flux was fine-tuned to match the flux of the titanium using shuttered RHEED intensity oscillations. Due to the imperfect growth control, the exact thickness of these films is determined to be slightly larger than 8 u.c. with incomplete surface layers, as discussed in Supplementary Note 7.

**Crystal truncation rods measurements and coherent Bragg rods analysis**. Crystal truncation rods (CTRs) were measured using a surface X-ray diffraction geometry with a six-circle diffractometer under X-ray photon energy of 16 and 23.9 keV at sectors 12-ID-D and 33-ID-D at Advanced Photon Source, Argonne National Laboratory. Both beamlines have a similar total flux of ~$2.0 \times 10^{12}$ photons s$^{-1}$. At 33-ID-D, the X-ray beam was focused by a pair of Kirkpatrick–Baez mirrors down to a beam profile of 50µm (vertical) × 500µm (horizontal). The two-dimensional diffraction images of CTRs at each $L$ step in the reciprocal space were recorded with a pixel array area detector (Dectris PILATUS 100 K). Samples were protected under dry helium gas flow in a concealed sample cell during room temperature measurements. Low temperature measurement was achieved with a closed-cycle-cryocooler system (Advanced Research System Model DE-204). A large set of CTRs in the reciprocal lattice coordinate were measured for all three epitaxial CaTiO$_3$ films at both room temperature and low temperature (30 K), with $H_{max}$, $K_{max} = 8$ r.l.u., $L_{max} = 9$ r.l.u. under $2 \times 2 \times 2$ pseudocubic notation.

3D electron densities (EDs) for the complete atomic structures were reconstructed from the complete set of CTRs by using an iterative phase retrieval technique, known as coherent Bragg rods analysis (COBRA)[32,34,35,52,55], through an in-house developed MATLAB code, generalized for systems with symmetry lower than $4mm$ (or simple four-fold symmetry system). Experimental CTR data were first background subtracted, and then properly corrected for geometric and polarization factors. Initial atomic structural model was constructed based on bulk structures with fitted CaTiO$_3$ $c$ lattice constant using GenX software[56]. Within each iteration, real space and reciprocal space constraints are applied to reconstruct phase information from measured CTRs. The structural results yielded by the COBRA iterations are found to be insensitive to the initial model, as illustrated in Supplementary Note 8.

The generic approach for uncertainty analysis based on a parameterized model is not applicable to COBRA-generated EDs. A method called noise analysis based on statistical analysis is previously used to determine the uncertainties in COBRA results[32]. This method requires COBRA reconstructed EDs of a large number of groups of CTRs adding afterward with random noise and analyzes the degree of scatters in the interested values extracted from EDs, which is extremely costly for analyzing six different structures (three epitaxial structures at room temperature and 30 K) presented in this work. By taking advantages of ultrathin CaTiO$_3$ films and well-known substrates used in this work, we estimate the uncertainty by including 11 pseudocubic u.c. of substrates into the reconstructed EDs. The structures of substrate unit cells buried underneath the 5th u.c. away from the interfaces should maintain their bulk structures. Therefore, the deviation between EDs and bulk structures for the first 6 u.c. of substrates is used for estimating the magnitude of deviation between COBRA results and the true values. A comparison between above two methods is detailed in Supplementary Note 9.

**Density functional theory calculations**. The density functional theory (DFT) calculations use the plane-wave basis and projector augmented wave (PAW)[57] method within the Vienna Ab initio Simulation Package (VASP)[58]. The choice of the exchange-correlational functional is the Perdew–Burke–Ernzerhoff (PBE)[59] generalized gradient functional. Convergence tests indicate that energies are converged to within 1 meV atom$^{-1}$ with a 560 eV cutoff energy, with 20 Å of vacuum in the direction perpendicular to the interface, and with a $8 \times 8 \times 1$ **k**-point mesh. Spin-polarization is used in all the calculations. Structure relaxation is iterated until the energy differences are below $10^{-6}$eV and until all forces on the atoms are below 0.05 eV Å$^{-1}$. To minimize the unphysical dipole energy arising from the heterostructure, CaTiO$_3$ thin films were symmetrically introduced on both side of the substrate. During the calculation, all the substrate atoms are fixed at their initial positions and are not allowed to relax.

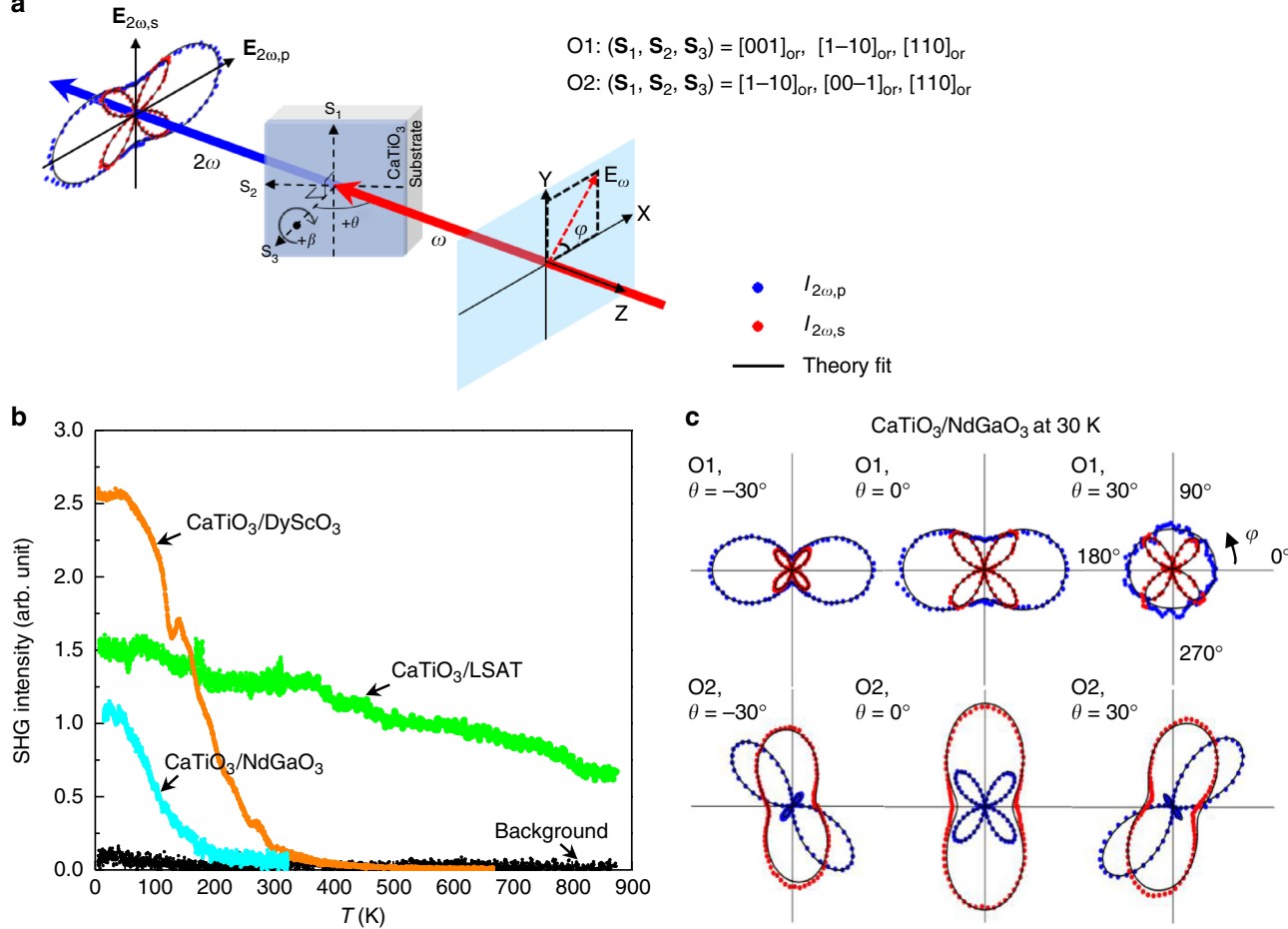

**Fig. 5** SHG measurements on ultrathin CaTiO$_3$ films. **a** Schematic of far-field transmission optical second harmonic generation (SHG) setup. Linear polarized fundamental $\lambda = 800$ nm, with polarization direction described by $\varphi$ is incident onto the sample at an angle $\theta$. Transmitted s-/p-polarized SHG signal $\mathbf{E}_{2\omega,s}$, $\mathbf{E}_{2\omega,p}$ at $\lambda = 400$ nm is measured. **b** Temperature dependent SHG signal of CaTiO$_3$ on NdGaO$_3$, DyScO$_3$, and LSAT reveals Curie temperature $T_C$ of 200, 350, and >900 K, respectively. **c** Polarimetry signal $I_{2\omega,s}$ (red dots), $I_{2\omega,p}$ (blue dots) on CaTiO$_3$/NdGaO$_3$ at 30 K are shown. Two sample orientations, O1 and O2, are experimentally measured as described by ($\mathbf{S}_1$, $\mathbf{S}_2$, $\mathbf{S}_3$) under orthorhombic notation. Theory fits (black lines) reveals a single domain $m$ symmetry of the CaTiO$_3$ on NdGaO$_3$ at 30 K

**Scanning transmission electron microscopy imaging.** The scanning transmission electron microscopy (STEM) images of CaTiO$_3$ thin films on NdGaO$_3$, DyScO$_3$, and LSAT under $(100)_{pc}$, $(010)_{pc}$, and $(110)_{pc}$ zone axes were collected on an FEI Titan G2 double aberration-corrected HR-STEM at 300 kV with a probe illumination angle of 28 mrad. High-angle annular dark-field (HAADF) and annular bright field (ABF)-like images were obtained with collection angles of 42–244 and 9–51 mrad, respectively. At each sample location, images were taken with the STEM fast scan direction set to 0° and 90° with respect to the substrate interface direction. These image pairs were then drift corrected, after which the images were superimposed.

Analysis of the STEM images was performed using custom-written MATLAB code. The sub-pixel resolution of the cations and oxygen positions were determined by fitting a seven parameter 2D elliptical Gaussian profile (to account for any ellipticity in the intensity distribution) to the HAADF/ABF intensity distribution.

**Optical second harmonic generation measurements.** Optical second harmonic generation (SHG) polarimetry and temperature dependent measurements were performed in a far-field transmission setup using femtosecond pulses at $\lambda = 800$ nm generated by a Spectra-Physics Empower Q-switched Nd:YLF pumped Solstice Ace Ti:Sapphire laser system (100 fs, 1 kHz). The experimental schematic is shown in Fig. 5a, where a linear polarized fundamental field with polarization direction $\varphi$ incident on the sample at an incident angle $\theta$. The p-polarized ($I_{2\omega,p}$) and s-polarized ($I_{2\omega,s}$) components of second harmonic field ($\mathbf{E}_{2\omega}$) was first spectrally filtered and then detected by a photo-multiplier tube, using lock-in method (SR830). For each sample, systematic polarimetry was performed by rotating the incident polarization $\varphi$ at fixed $\theta$ for two different sample orientations (O1 and O2). SHG polarimetry on CaTiO$_3$/NdGaO$_3$ at 30 K was performed at incident angles $\theta = -30°$, 0°, 30° for two sample orientations, O1: ($\mathbf{S}_1$, $\mathbf{S}_2$, $\mathbf{S}_3$) = ([001]$_{or}$, [1−10]$_{or}$, [110]$_{or}$), O2: ($\mathbf{S}_1$, $\mathbf{S}_2$, $\mathbf{S}_3$) = ([1−10]$_{or}$, [00−1]$_{or}$, [110]$_{or}$), as shown in Fig. 5c (red and blue dots). Similar SHG behavior are also observed in CaTiO$_3$/

DyScO$_3$ at 30 K (See Supplementary Figure 10). SHG polarimetry on CaTiO$_3$/LSAT with sample orientations, O1: ($\mathbf{S}_1$, $\mathbf{S}_2$, $\mathbf{S}_3$) = ([100]$_{pc}$, [010]$_{pc}$, [001]$_{pc}$), O2: ($\mathbf{S}_1$, $\mathbf{S}_2$, $\mathbf{S}_3$) = ([010]$_{pc}$, [−100]$_{pc}$, [001]$_{pc}$), reveals a similar pattern at 30 K and room temperature, as shown in Supplementary Figure 10, and is different from CaTiO$_3$ on NdGaO$_3$ and DyScO$_3$. Temperature-dependent measurements were performed by monitoring the SHG signal while scanning the sample temperature, which was controlled using helium cooled Janis 300 cryostat (for low temperature) and a heater (for high temperature).

Symmetry analysis of the SHG polarimetry was performed using an analytical model described below[60,61]. Fundamental field is written as $(E_\omega\cos(\varphi), E_\omega\sin(\varphi), 0)$ under the laboratory coordinates $(X, Y, Z)$, and incident onto sample at an angle $\theta$. Sample orientation can be described by $\beta$, with $\beta = 0°$ for O1 and $\beta = 90°$ for O2. Considering refraction and transmission at sample surface, the fundamental field $E'_{\omega,i}$ inside the sample can be expressed as

$$E'_{\omega,1} = (\cos(\theta')\cos(\beta)\cos(\varphi)t_p - \sin(\beta)\sin(\varphi)t_s)E_\omega \quad (1)$$

$$E'_{\omega,2} = (\cos(\theta')\sin(\beta)\cos(\varphi)t_p + \cos(\beta)\sin(\varphi)t_s)E_\omega \quad (2)$$

$$E'_{\omega,3} = -\sin(\theta')\cos(\varphi)t_pE_\omega \quad (3)$$

Where $\sin(\theta') = \sin(\theta)/n$, $n$ is refractive index, and $t_p = 2\cos(\theta)/[n\cos(\theta)+\cos(\theta')]$ and $t_s = 2\cos(\theta)/[\cos(\theta) + n\cos(\theta')]$ are Fresnel coefficients. The SHG field $E'_{2\omega,i}$ generated inside the sample can be calculated by $E'_{2\omega,i} = d_{ijk}E'_{\omega,j}E'_{\omega,k}$, $d_{ijk}$ is nonlinear SHG coefficients, or under Voigt notation, $E'_{2\omega,i} = d_{ij}E'^{,Voigt}_{\omega,j}$. $d_{ij}$ matrix

for $m$ point group symmetry is:

$$d^m = \begin{pmatrix} 0 & 0 & 0 & 0 & d_{15} & d_{16} \\ d_{21} & d_{22} & d_{23} & d_{24} & 0 & 0 \\ d_{31} & d_{32} & d_{33} & d_{34} & 0 & 0 \end{pmatrix} \qquad (4)$$

To simplify the analysis, we ignored the dispersion effect, i.e., $n = n_\omega \approx n_{2\omega}$. The transmitted SHG field in the laboratory coordinates is:

$$E_{2\omega,p} = E_{2\omega,X} = (\cos(\theta')\cos(\beta)E'_{2\omega,1} + \cos(\theta')\sin(\beta)E'_{2\omega,2} - \sin(\theta')E'_{2\omega,3})t'_p \quad (5)$$

$$E_{2\omega,s} = E_{2\omega,Y} = (-\sin(\beta)E'_{2\omega,1} + \cos(\beta)E'_{2\omega,2})t'_s \qquad (6)$$

Where $t'_p = 2n\cos(\theta')/[n\cos(\theta) + \cos(\theta')]$, $t'_s = 2n\cos(\theta)/[\cos(\theta) + n\cos(\theta')]$. SHG intensity from sample is $I_{2\omega,p} = \alpha|E_{2\omega,p}|^2$ and $I_{2\omega,s} = \alpha|E_{2\omega,s}|^2$, where $\alpha$ is a constant. In above equations, $\alpha$ and $E_\omega$ can be further eliminated by defining effective SHG matrices as $d_{ij}^{eff} = \sqrt{\alpha}E_\omega^2 d_{ij}$. For CaTiO$_3$ on NdGaO$_3$ and DyScO$_3$, single domain state of CaTiO$_3$ films is observed. Explicitly, we have following equations for orientations O1 and O2:

$$\text{O1}: \begin{cases} I_{2\omega,p}^{total} = I_{2\omega,p}(\beta = 0°) \\ I_{2\omega,s}^{total} = I_{2\omega,s}(\beta = 0°) \end{cases} \qquad (7)$$

$$\text{O2}: \begin{cases} I_{2\omega,p}^{total} = I_{2\omega,p}(\beta = 90°) \\ I_{2\omega,s}^{total} = I_{2\omega,s}(\beta = 90°) \end{cases} \qquad (8)$$

For CaTiO$_3$ on LSAT, four equivalent domains represented by different $\beta$ values are considered. Under the phase uncorrelated approximation, we have following equations:

$$\text{O1}: \begin{cases} I_{2\omega,p}^{total} = w_1 I_{2\omega,p}(\beta = 0°) + w_2 I_{2\omega,p}(\beta = 90°) + w_3 I_{2\omega,p}(\beta = 180°) + (1 - w_1 - w_2 - w_3)I_{2\omega,p}(\beta = 270°) \\ I_{2\omega,s}^{total} = w_1 I_{2\omega,s}(\beta = 0°) + w_2 I_{2\omega,s}(\beta = 90°) + w_3 I_{2\omega,s}(\beta = 180°) + (1 - w_1 - w_2 - w_3)I_{2\omega,s}(\beta = 270°) \end{cases}$$
(9)

$$\text{O2}: \begin{cases} I_{2\omega,p}^{total} = w_1 I_{2\omega,p}(\beta = 90°) + w_2 I_{2\omega,p}(\beta = 180°) + w_3 I_{2\omega,p}(\beta = 270°) + (1 - w_1 - w_2 - w_3)I_{2\omega,p}(\beta = 0°) \\ I_{2\omega,s}^{total} = w_1 I_{2\omega,s}(\beta = 90°) + w_2 I_{2\omega,s}(\beta = 180°) + w_3 I_{2\omega,s}(\beta = 270°) + (1 - w_1 - w_2 - w_3)I_{2\omega,s}(\beta = 0°) \end{cases}$$
(10)

Where $w_1$, $w_2$, $w_3$, are the area fraction of three of the four domain variants in the probed area. The fits reveal these factors to be ~1/4 each as suggested by COBRA results.

**Code availability**. The computer codes that support the findings of this study are available from the corresponding author upon reasonable request.

## Data availability

The data that support the findings of this study are available from the corresponding author upon reasonable request.

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

## Acknowledgements
This work was supported by National Science Foundation (NSF) MRSEC Center for Nanoscale Science under award no. DMR-1420620. This research used resources of the Advanced Photon Source, a U.S. Department of Energy (DOE) Office of Science User Facility operated by Argonne National Laboratory under Contract No. DE-AC02-06CH11357. The DFT study is conducted as part of The Pennsylvania State University (PSU) 2D Crystal Consortium – Materials Innovation Platform (2DCC-MIP) under NSF award No. DMR-1539916. Support was also received from the PSU's Institute for CyberScience (ICS) through a Graduate Research Assistantship. Computations for this research were performed on the PSU's Institute for CyberScience Advanced CyberIn-frastructure (ICS-ACI). This content is solely the responsibility of the authors and does not necessarily represent the views of the ICS. This work made use of the Cornell Center for Materials Research (CCMR) Shared Facilities, which are supported through the NSF MRSEC program (DMR-1719875). Substrate preparation was performed in part at the Cornell NanoScale Science & Technology Facility, a member of the National Nano-technology Coordinated Infrastructure (NNCI), which is supported by the DSF ECCS-1542081. We thank Zhan Zhang for the help with synchrotron XRD at APS-33ID and Haiying Wang from the Materials Characterization Laboratory at Penn State for the help with STEM sample preparation.

## Author contributions
Y.Y. and H.Z. performed the synchrotron CTR measurements and COBRA analysis. Y.L. and S.B.S. formulated the model and performed DFT calculations. K.W., Y.Y., and G.S. carried out scanning transmission electron microscopy studies. C.M.B. synthesized the samples with advice from D.G.S. Y.Y. and V.G. performed optical SHG measurements. Y. Y., D.G.S., S.B.S., H.Z., and V.G. prepared the manuscript. All authors discussed the results.

## Additional information

**Competing interests:** The authors declare no competing interests.

