## [Peer Review File · Nature Communications]

Reviewers' comments:

Reviewer #1 (Remarks to the Author):

The authors apply an X-ray phase retrieval technique to determine the atomic structures of epitaxial CaTiO₃ films grown on low symmetry substrates revealing interfacial coupling of the oxygen octahedra and an unexpected out-of-plane polarization in the thin CTO films. The structural results are found to be in agreement with first principles calculations and TEM results demonstrating the efficacy of the non-destructive X-ray 3D imaging technique they apply to study ultra-thin films. These results will be of general interest to the field of oxide interfaces however, significant issues discussed below should be addressed before I can recommend this manuscript for publication.

Based on the experimental error bars in Figure 3 and the DFT results, the most significant changes in the octahedral rotations of the films occurs within the first 2-3 unit cells (~4uc for the CTO/LSAT) film. It is not clear to me how the distortions in the interfacial CTO layers will lead to the out-of-plane polarization of the entire 8 uc thick film. Could the authors comment on why this is the case? Are there any insights from DFT?

My concern here is that, within the experimental errors, the value of the polarization for the CTO film on LSAT (Figure 3) is comparable to the top layers of the LSAT substrate (~10 microC/cm²) which is expected to be paraelectric.

What are the amplitudes of the displacements in nm? Also, how is the sign of the in-plane polarization for the CTO on LSAT determined if you are averaging over 4 rotational domains?

The procedure used to determine the uncertainty in the atomic position used here (Zhou et al. J Phys D 45) shows that the error bars for the surface layers can be as large as a factor of 2 compared to the bulk layers included in the analysis. Do the authors check that the error in the surface positions determined from the COBRA analysis is the same as the bulk layers before deciding to only use the layer in the bulk errors for the entire analysis?

Ferroelectricity in CTO films on LSAT and NGO has been observed in reference 43 for example with a T_c~170 K. In ref 43, a different rotational pattern, a+a+c0) for the CTO films on LSAT was observed. Was this considered by the authors by measuring half order Bragg peaks associated with the CTO film?

The authors state in the summary on page 3 that they 'employ COBRA' to 'develop a phase-retrieval algorithm..' which may be misleading. The COBRA- phase retrieval algorithm is applied to low-symmetry systems in the current manuscript but has been previously applied to other systems in references 20, 31-41 and the current manuscript represents a specific application of the technique.

Technical questions

1. Does the starting model of the films for the COBRA analysis include rotations of the octahedra in the films and if so, how sensitive are the final extracted angles to initial model and the Gaussian fits?
2. Are the polar distortions resolvable in the TEM images? If so, how do they compare with theory and the COBRA measurements?
3. The DFT results for the films in Figure 3 a-c stop at 6 unit cells, however, the caption for Figures 3d-e and the description on page 13 refers to calculations for 8 unit cells? Is this correct?
4. What are the strain states of the films? Can the authors include the out-of-plane lattice spacings determined from the phase-retrieval analysis?
5. Presumably, the film surface terminations will be different since the substrates are either AO or BO₂ terminated. Is this confirmed by COBRA and/or TEM?

Minor Corrections

1. What are the units of L in Figure 1(d)? Are they in orthorhombic NGO units or the 2x2x2 pseudocubic unit cell? If so, that should be defined in the caption and tick marks included on the horizontal axis to identify the peaks.
2. Could the authors clarify whether the experimental angular values in Supplementary Table 2 are averaged over the entire film?
3. Supplementary note 1 and Supplementary Figure 1 should state the H,K,L values are in terms of the 2x2x2 pseudocubic unit cell.

Reviewer #2 (Remarks to the Author):

In the manuscript NCOMMS-18-16565, the authors study three ultrathin CaTiO₃ films on different substrates and reconstruct 3D oxygen tilts and polar displacements using coherent Bragg rod analysis. Atomic imaging of those thin films using Cs-STEM and DFT results were also performed to give understanding to the observations. The authors observe the highest polar to non-polar transition (900 K) in their ultrathin CaTiO₃/LSAT film as compared to reported thicker one (140-190 K); they reveal an unexpected out-of-plane polar displacement; they also explored the intertwining roles of tilt epitaxy, substrate strain, and substrate surface terminations. The experimental aspects of this work are outstanding and essential for moving forward our general understanding of tilt epitaxy engineering in ultrathin perovskite films. Overall, I think this is a very novel finding that will be of broad interest.

(1) In studies such as this, it is nice to know the presence of switchable P-E loops in those extraordinary ultrathin CaTiO₃ films. In particular, the authors term ferroelectric transition in the manuscript. If the polarity cannot be switched, the system is just polar. I think that the switching characteristics have to be provided if the authors claim a ferroelectric phase and possible applications.

(2) The authors reveal the unexpected out-of-plane polar displacement in ultrathin CaTiO₃ films and consider it as interfacial valence mismatch effect. On the other hand, the depolarization field may suppress the out-of-plane ferroelectricity in their ultrathin films as well. The authors need to provide more detailed discussion on this issue.

(3) In epitaxial heterojunctions, the film-substrate misfit strains often can be relaxed by the crystallographic defects such as dislocations with typically few hundred nanometers or be accommodated by uniform deformations of unit cells. With a relatively broad probe area of X-ray (compared to Cs-STEM), those imperfections may significantly affect the Bragg peaks and not been considered in the phase retrieval algorithm fitness. As stated by the authors, they deal with the most complex low symmetry and bulky structural reconstruction to date by the technique. The authors may briefly comment on this concern.

(4) There is no details of "Methods" and DFT calculations in the main text and supplementary information.

(5) The authors may recheck references more carefully. For example pages are missing in ref 41 and 43. On the other hand, the authors cited a few theoretical works about the octahedral rotation-induced ferroelectricity while experimental demonstrations of switchable improper mode involving octahedral rotations are well recognized recently, e.g. Bousquet et al., Nature 452, 732 (2008) and Oh et al., Nature Materials 14, 407 (2015).

Reviewer #3 (Remarks to the Author):

June 22, 2018

Brief summary of the manuscript:

The authors present a detailed atomic structural study investigating substrate-induced octahedral tilt patterns in ultra-thin films of CaTiO₃ on three different substrates with differing symmetries and lattice mismatches. The main results are drawn from a comprehensive 3-dimensional analysis of the electron density reconstruction over the whole film thickness, including the top layers of the substrate material, using the iterative phase retrieval method COBRA. The established COBRA method has been adapted by the authors to work also on systems with a low in-plane symmetry, and the structural results obtained from the 3D electron density maps are corroborated by complementary experimental techniques, particularly STEM and optical SHG, as well as by comparison to density functional theory.

General remarks:

The novelty of the described research is mostly technical in nature: The reported results certainly represent some of the most complete atomic structural determinations of thin film systems, particularly when taking the low symmetry of these systems into account. The structural insights gained from these measurements can help to disentangle the individual contributions of surface termination and symmetry, lattice strain, and structural distortions across the interface caused by a continuous change in the oxygen octahedral network orientation and tilt patterns. Thus, the method provides crucial experimental input for a basic understanding of the response of these thin film systems subjected to different substrate constraints, which should ultimately allow for a tailored design of specific desired film properties.

The observed effects in this study are not surprising or all new by themselves, and the concept of induced or coherent structural distortions across interfaces in thin film or multilayer systems has been around for decades. I would thus encourage the authors to devote an additional paragraph in the introductory part of the manuscript to specifically review the different cases that have been discussed in the literature thus far, in particular with regards to the octahedral tilt patterns. A number of the relevant papers are cited in the introduction in "bulk form" (17-26), but their relation and relevance with respect to the present study is not discussed.

From a technical point of view, the manuscript is very solid and well written, although it should be revised for minor grammatical and language issues. Figures are clear and illustrate well the contents described in the text. The relevant literature is adequately cited. Conclusions and interpretations of the structural results are well supported by experimental results (note that I am not an expert in DFT and SHG measurements, so I am unable to assess those corresponding results and interpretations in detail).

My main concern with the manuscript is related to the novelty, significance, and impact of the reported findings in the scientific community and in relation to the publication requirements by Nature Communications. As detailed above, the technical achievement in obtaining 3D ED maps at this level of detail for low-symmetry systems is undisputed. However, the method is in principle based on a technique (COBRA) which has been around for 15 years (a good number of citations to earlier works are cited in the manuscript), and which has been further refined in this work (generalization to lower symmetry systems) to obtain the presented results. Although a number of excellent structural studies have been published with COBRA since its inception, its broader impact has been limited so far. To be completely honest, I am not fully convinced that the present manuscript will change this situation significantly, although the authors state in their conclusions that "With the development of high-energy surface X-ray diffraction that enables the rapid capture of large portions of 3D Bragg rods in reciprocal space, COBRA measurements will become much more efficient and widespread in their application to oxide thin films and heterostructures". I would also argue that the last part of the last sentence, "This work will motivate progress ..., and more broadly, provide a powerful non-destructive tool with atomic resolution for probing the electron density of complex functional interfaces.", is somewhat exaggerated in this context. In my view, this work rather provides an important and very beneficial improvement to a tool which has, however, been available since long before the present work. I would therefore ask the authors to

clarify these statements and substantiate their claim why the technique would suddenly gain so much in importance through the work presented in this manuscript. Note again that I am not disputing the validity of the presented results, I am merely questioning the impact of this particular work with regards to making the experimental technique itself more accessible to a broader community working on complex functional interfaces and beyond.

In light of the above concerns, I cannot recommend the manuscript as is for publication in Nature Communications for the sole reason that I don't think the manuscript fulfills the requirements for novelty, significance for the field and interest of a broad audience. Since the manuscript is of very high quality in all other aspects, I recommend that the authors either attempt to better justify the expected broad impact of their paper, or submit to a more specialized journal in the field of complex functional oxide systems or thin film structure determinations.

Miscellaneous issues and comments:

1) Figure 2: This is a beautiful figure illustrating the 3D data obtained from the COBRA method. However, a few of the important details are hard to capture simply due to the fact that the corresponding graphics are very small. For example, the pie charts indicating the octahedral rotation angles next to the 3D representation of the ED map are very hard to discern, particularly on a printed piece of paper. The same is true for the insets highlighting, for example, the dashed elliptical lines outlining the oxygen atom positions. These lines are barely visible, and their dashed nature is impossible to see. I would suggest to enlarge these important details if possible.

Reviewers' comments:

We thank the reviewers for their detailed comments. Their efforts and interest are greatly appreciated. We have taken their comments to heart, and over the last 3 months, we worked diligently to address each comment in as detailed manner as possible. This includes extensive new simulations and analysis, which has resulted in 8 new supplementary figures and 5 supplementary notes.

Reviewer #1 (Remarks to the Author):

The authors apply an X-ray phase retrieval technique to determine the atomic structures of epitaxial CaTiO₃ films grown on low symmetry substrates revealing interfacial coupling of the oxygen octahedra and an unexpected out-of-plane polarization in the thin CTO films. The structural results are found to be in agreement with first principles calculations and TEM results demonstrating the efficacy of the non-destructive X-ray 3D imaging technique they apply to study ultra-thin films. These results will be of general interest to the field of oxide interfaces however, significant issues discussed below should be addressed before I can recommend this manuscript for publication.

We are very glad that the reviewer thinks that our results will be of general interest to the field of oxide interfaces. And we thank the reviewer for these detailed expert comments.

1. Based on the experimental error bars in Figure 3 and the DFT results, the most significant changes in the octahedral rotations of the films occurs within the first 2-3 unit cells (~4uc for the CTO/LSAT) film. It is not clear to me how the distortions in the interfacial CTO layers will lead to the out-of-plane polarization of the entire 8 uc thick film. Could the authors comment on why this is the case? Are there any insights from DFT?

This is an excellent observation. Both experiments and DFT confirm this “**Domino**” effect, where a larger structural distortion (octahedral tilt changes) within the first few unit cells of the film has a knock-on effect on both subsequent tilts and the emergence of out of plane polarization through the rest of the film thickness of 8-unit cells.

First, we note that the electrostatic interactions are long range. A significant out of plane polarization in the first few layers can create knock-on out-of-plane polarization in the rest of the film, and eventually compensate the surface depolarization field with external and internal compensating charges. Structural distortions (e.g. polar distortion) are also typically long range due to strain and octahedral tilt compatibility conditions.

We note that:

(1) The measured out-of-plane polarization decreases away from the film interface to the surface as seen in main **Figs. 3a-c** from both COBRA and DFT results.

(2) The corresponding structural distortions induced by tilt epitaxy (the three components of oxygen tilts, replotted below for clarity) are *finite* within the full thickness of the ultrathin film, even as they decay with distance away from the interface. This may be obvious for some tilt angles seen below (**Fig. R1**), but somewhat subtle for others, such as CTO/NGO, where the average tilt value (within the error bar) is slightly above the mean bulk value for the tilt. In a follow-up response to another question below, we show that our error analysis is more conservative than the calculated error by a different method from literature. Hence, we are confident that these mean values of tilts are robust and reflect finite values through the film thickness.

We now modify the main text to address this question and our response above.

Figure R1 | Tilt angles **a** α , **b** β , **c** γ for CTO films on NGO (green), DSO (blue), and LSAT (red) substrates. COBRA and DFT results are plotted using dots and solid lines respectively. Black dashed lines give bulk tilt angles for reference.

2. My concern here is that, within the experimental errors, the value of the polarization for the CTO film on LSAT (Figure 3) is comparable to the top layers of the LSAT substrate ($\sim 10 \mu\text{C}/\text{cm}^2$) which is expected to be paraelectric.

Considering our error analysis tends to be conservative (as discussed in detail below), the reconstruction might appear to show some mean polarization value near the surface of LSAT ($\sim 5 \pm 5 \mu\text{C}/\text{cm}^2$), which could be due to the influence from the film-substrate interface. We further noted that, the CTO polarizations ($10\text{--}15 \mu\text{C}/\text{cm}^2$) in main **Fig.3c** are still clearly larger than the error bar ($\leq 5 \mu\text{C}/\text{cm}^2$), which indeed reflect a robust polarization. Also, the polarization values do increase significantly from the LSAT substrate to the film, as can be seen in **Fig.3c**.

To gain further confidence in these results, we performed optical SHG measurements, that CTO/LSAT does show clear signal while the substrate signal is close to noise, as shown in **Fig. 5b**. Optical SHG is a very sensitive tool to probe inversion symmetry breaking even when the polar displacements are very small, and this clearly suggests that a polar state is indeed stabilized in the CTO on LSAT and provides confidence in our COBRA results.

3. What are the amplitudes of the displacements in nm? Also, how is the sign of the in-plane polarization for the CTO on LSAT determined if you are averaging over 4 rotational domains?

The displacements extracted from the COBRA results are calculated based on the center-of-mass of A-site, B-site and oxygen atoms in each pseudocubic unit cell, while the center-of-mass of oxygens is chosen to be the reference position. Hence, A-site and B-site cations displacement can be plotted as shown in **Fig. R2** below.

Since four in-plane domains of CTO on LSAT are symmetrically equivalent, there is freedom in plotting the in-plane polarization directions. Here, we plot the positive polarizations along a-axis to directly compare with the DFT results shown in main **Fig. 3c**. This is not uniquely determined from the COBRA analysis due to the domain folding.

We emphasize this detail in the revised main text.

Figure R2 | Cations displacements in **a** CTO/NGO **b** CTO/DSO **c** CTO/LSAT at 30 K.

4. The procedure used to determine the uncertainty in the atomic position used here (Zhou et al. J Phys D 45) shows that the error bars for the surface layers can be as large as a factor of 2 compared to the bulk layers included in the analysis. Do the authors check that the error in the surface positions determined from the COBRA analysis is the same as the bulk layers before deciding to only use the layer in the bulk errors for the entire analysis?

This is an important comment regarding the uncertainty estimation of COBRA method.

To summarize, our method of error analysis (as detailed below) is more conservative than Zhou's method (also described below).

We demonstrate this by comparing the uncertainties given by the above two methods for the case of CTO/NGO at 30 K. We first construct 8 sets of CTR data (**Fig. R3**) with simulated random noise adopting the method proposed by Zhou et al. (J. Phys. D: Appl. Phys. 45 (2012) 195302), as shown in the plot below for (0 4) rod.

Figure R3 | (0 4) crystal truncation rod of CTO/NGO with 8 different sets of simulated noise are plotted as an example.

The COBRA results on these 8 sets of CTR data are plotted in the figure below (**Fig. R4**), where blue dots and their error bars are the results from the original experimental data and simplified uncertainties estimation proposed by this work. The red open circles are results based on 8 sets of CTR data with simulated noise. The grey area presents the statistical uncertainties of all the 8 data sets.

We can conclude that the uncertainties given by our simplified method provide an **excellent conservative** estimation of the upper limits of uncertainty in the COBRA results.

Figure R4 | Uncertainty analysis on CTO/NGO at 30 K based on Zhou's bootstrap approach is plotted for tilt angles and polarizations using red open circles. The grey area represents the uncertainty determined

from the scattering of the red open circles. The blue dots with error bars are original results using a simplified uncertainty estimation as described in the method section of the current manuscript. The blue error bars are close to the upper limit of uncertainties from the bootstrap approach throughout the entire thickness.

Additional summary note on Zhou's bootstrap approach versus our approach: Since COBRA uses an iterative algorithm to reconstruct the real space electron density, traditional uncertainty analysis for fitting process, that based on a parameterized numerical model, is not applicable. Previously, Zhou et al, proposed a method based on the bootstrap approach to estimate the uncertainty of COBRA results. In their method, experimental CTR data are first randomly perturbed by simulated noise functions, and then analyzed by the COBRA method, yielding a new set of real space results. This process can be repeated for multiple times, which generates multiple sets of real space results. Finally, the uncertainties can be estimated by the scattering extent of these sets of results. This method is a systematic way of determining COBRA results' uncertainties, however, repeating COBRA iterations for a large number of times to gain statistics can be *extremely time and effort consuming*, especially for low symmetry systems of complex oxides, where a large unit cell and a large number of inequivalent atoms are involved. For example, in the cases of CTO/NGO and CTO/DSO presented by this work, positions of more than 350 atoms in 3D are needed for each analysis. Performing uncertainty analysis for all three systems at both room temperature and 30 K using the bootstrap approach can be formidable.

Here we propose a straightforward way of estimating uncertainty as described in the Method section of the manuscript. The deviation of atomic positions from bulk values in the first several unit cells in the substrate that are far beneath the substrate-film interfaces (> 5 u.c.) can be used as an estimation of uncertainties in COBRA analysis. Although this method gives an efficient way of estimating the uncertainty, it might not reflect the fact that uncertainties may vary as going from substrate to film surface. Our analysis above indicates that this method is more conservative than Zhou's bootstrap method.

Above discussion has been added to the revised supplementary materials.

5. Ferroelectricity in CTO films on LSAT and NGO has been observed in reference 43 for example with a $T_c \sim 170$ K. In ref 43, a different rotational pattern, $a+a+c_0$) for the CTO films on LSAT was observed. Was this considered by the authors by measuring half order Bragg peaks associated with the CTO film?

Yes, we have performed half order peak measurement for CTO on LSAT to determine the octahedral tilt pattern. According to our measurement, the LSAT substrate has cubic lattice parameters, but it contains small double perovskite structural domains, which give rise to very broad and strong substrate peaks at half order positions (H, K, L = half integers under pseudocubic notation), overwhelming the weak CTO film peaks associated with all out-of-phase tilts, as seen in **Fig. R5a**. Thus, direct determination of out-of-phase tilts is not possible.

However, half order peaks associated with in-phase tilts at $(\text{odd}/2, \text{even}/2, \text{odd}/2)$ or $(\text{even}/2, \text{odd}/2, \text{odd}/2)$ are observed as shown in **Figs. R5c** and **d**, suggesting in-phase tilts about a- and b-axes. Since CTO film on LSAT has four equivalent in-plane directions and we do not observe a double splitting peak feature, as contrary to the observations in ref. [43] (ref. [49] in revised manuscript), these peaks should be explained by multidomain states of a single in-phase tilt along a- or b-axis (not both).

We also confirm that the absence of $(1.5\ 0.5\ 4)$ peak (**Fig. R5b**), suggesting an out-of-phase or no tilt along c-axis. Hence, we conclude from the half order peak results that the CTO on LSAT only has one in-phase tilt along one of the in-plane directions, and could have out-of-phase tilts or no tilts along the other two axes.

However, COBRA results reveal a finite tilt along the c-axis, ruling out the possibility of c^0 . Thus, $a^+b^-c^-$ or $a^-b^+c^-$ tilt pattern is expected. Moreover, our DFT study suggests a tilt pattern of $a^-b^+c^-$ in CTO on

LSAT, and the most stable tilt pattern of CTO under a similar strain state (~1%) on NGO substrate is $a^+b^+c^-$. Hence we conclude that the CTO on LSAT should adopt an $a^+b^+c^-/a^+b^-c^-$ mixed tilt pattern.

These arguments are captured briefly in the main text and detailed in revised **Supplementary Note 3**.

Figure R5 | a Strong (1.5 0.5 2.5) peak from LSAT overwhelm the possible out-of-phase tilt from CTO thin film. **b** The absence of (1.5 0.5 4) peak indicates a c^- or c^0 tilt about c-axis. **c-d** The single peak at (1 0.5 3.5) and (0.5 1 3.5) position suggests the CTO on LSAT has only one in-phase tilt along one of the in-plane axes.

6. The authors state in the summary on page 3 that they ‘employ COBRA’ to ‘develop a phase-retrieval algorithm..’ which may be misleading. The COBRA- phase retrieval algorithm is applied to low-symmetry systems in the current manuscript but has been previously applies to other systems in references 20, 31-41 and the current manuscript represents a specific application of the technique.

This expression has been changed in revised manuscript to avoid confusion as described below.

‘We employ Coherent Bragg Rods Analysis (COBRA) which reconstructs 3D electron density with atomic resolution based on a phase retrieval algorithm taking advantage of the interference between the diffracted X-ray beams from the thin film and the substrate.’

Technical questions

7. Does the starting model of the films for the COBRA analysis include rotations of the octahedra in the films and if so, how sensitive are the final extracted angles to initial model and the Gaussian fits?

Yes, the starting model of the films has included oxygen octahedra tilts. The final extracted angles are not sensitive to the initial model and the fits. The analysis process is detailed as following and demonstrated by the analysis on the data of CTO/NGO at 30 K.

First, we construct an initial model of CTO film with correct tilt pattern ($a^-b^+c^-$). This can be determined by analyzing the half order peaks adopting the method discussed above. (see the response to the 5th comments of reviewer 1) As shown in **Fig. R6**, the combination of (1 2 5), (-1 -1 3), and (-1 3 3) peaks (under $2\times 2\times 2$ pseudocubic notation) from CTO clearly indicate a tilt pattern of $a^-b^+c^-$ for CTO film on NGO substrate.

Figure R6 | Half order peaks of CTO/NGO under $2\times 2\times 2$ pseudocubic notation. **a** The presence of (1 2 5) peak from CTO film indicates an in-phase tilt about b -axis (b^+). The asterisk (*) marks the NGO substrate peak position. **b** The (-1 -1 3) peak of CTO film suggests either a^- tilt or b^- tilt. Since the tilt about b -axis has been determined to be b^+ , the tilt about a -axis is out-of-phase (a^-). **c** Similarly, the (-1 3 3) peak from CTO suggests a c^- tilt about c -axis.

Then, we can choose different values for the three tilts angles of CTO. In this step, there is freedom in the choice of the angles. Usually, we can use bulk tilt angles of CTO as the starting point. Here, to demonstrate that the final results are not sensitive to the initial tilt angles, we used a different initial model with tilt angles of 7° , 5° , and 4° respectively for α , β , and γ . To overcome the stagnation during the COBRA iterations, we applied an atomicity constraint, which is to remove the unphysical features in the electron densities after every 10-20 iterations. The atomicity constraint was applied manually by constructing a new initial model for future iterations based on the atom positions yielded from previous iterations. [PRB 71, 144112 (2005)] This process can be repeated multiple times until the results converge. **Figure R7** shows the tilts angles of all the intermediate states during the analysis on CTO/NGO. The COBRA analysis started with the initial model of 7° , 5° , and 4° respectively for α , β , and γ , producing an intermediate tilt state shown by 'Result 1' in **Fig. R7**. Then, we parameterized the atoms positions from the electron density of 'Result 1' and constructed a new starting model for the next 10-20 iterations, which produced 'Result 2', etc. After repeating this process for four times, the analysis converged to 'Result 4'. As we can see in **Fig. R7**, the tilts angles yielded by this result are very close to the results presented in main Fig. 3a, which is labeled as 'Final' in **Fig. R7**.

These COBRA results can also be partially complemented by STEM study. As shown in main **Fig. 4**, the COBRA yielded β angles are consistent with the STEM results. This further supports the validity of the COBRA analysis.

The details of this procedure are now captured in the supplementary section.

Figure R7 | **a** α , **b** β , and **c** γ values for all the intermediate states during the COBRA analysis. The atomicity constraint was applied every 10-20 COBRA iterations. The tilts angles given by ‘Result 4’ is considerably close to the results presented in the main text (labeled as ‘Final’).

8. Are the polar distortions resolvable in the TEM images? If so, how do they compare with theory and the COBRA measurements?

The STEM measurement in this work was performed at room temperature. At room temperature, only the CTO film on LSAT displays significant polarization, which is demonstrated by the COBRA results in **Supplementary Fig. 6** and temperature dependent SHG results in **Fig. 5b**. However, we found that the polarizations extracted from STEM at room temperature were quite noisy. As shown in **Fig. R8**, the error bars of the polarization extracted from STEM on CTO/LSAT are quite significant when compared to the mean values (solid lines). However, these mean values from STEM agree well with room temperature COBRA results (dots), suggesting non-zero polarizations in CTO along in-plane and out-of-plane directions. The fact that the COBRA error bar is smaller than that of STEM is probably due to the fact that X-ray diffraction employed in COBRA is macroscale in nature and the averaged structure information is being extracted over a large sample area.

Figure R8 | **a** In-plane and **b** out-of-plane polarization in CTO/LSAT at room temperature probed by STEM (solid lines) and COBRA (dots).

This discussion is now captured in the revised **Supplementary Note 4**.

9. The DFT results for the films in Figure 3 a-c stop at 6 unit cells, however, the caption for Figures 3d-e and the description on page 13 refers to calculations for 8 unit cells? Is this correct?

We thank the reviewer for pointing out this mistake. In DFT calculations, in order to rule out the artificial dipole interaction arising from the asymmetric supercell, we constructed a three-layer heterostructure with two CTO films sandwiching the substrate in the middle. To reduce the size of the systems, we used 6 u.c. thickness for the CTO films, instead of 8 u.c. Thus, the actual thickness in DFT calculations is 6 u.c.. Accordingly, we have revised caption in **Figs. 3d-e** and description in main text.

10. What are the strain states of the films? Can the authors include the out-of-plane lattice spacings determined from the phase-retrieval analysis?

The strain states for CTO films on NGO, DSO, and LSAT substrates are 1.1%, 3.3%, and 1.2% tensile, respectively. The out-of-plane lattice spacings from COBRA analysis are plotted in **Fig. R9** below. The black lines provide a guidance to eye, which represent the out-of-plane lattice spacings of bulk substrates and the corresponding lattice spacing for strained CTO films estimated using Poisson ratio ($\nu=0.174$) are shown as circles. The Poisson ratio is calculated by using the method in [*ACS Nano*, 2018, **12** (2), pp 1306–1312] and the strained CTO lattice parameters reported in [*Adv. Funct. Mater.* 2016, **26**, 7271–7279]. As shown in **Fig. R9**, the lattice spacings of these CTO films match well with the expected values, indicating the films are under fully strained states. Moreover, in the crystal truncation rods (CTRs) measurements of these films, all the CTRs are of high quality and nicely located at integer in-plane (H, K) reciprocal positions dictated by the substrates, which also suggests the fully strained states of these films.

Figure R9 | Out-of-plane lattice spacing of **a** CTO/NGO, **b** CTO/DSO, and **c** CTO/LSAT determined by COBRA. The black lines are values of bulk substrates and strain CTO estimated using Poisson ratio.

11. Presumably, the film surface terminations will be different since the substrates are either AO or BO₂ terminated. Is this confirmed by COBRA and/or TEM?

The CTO termination of the three systems can be seen clearly in the (110) pseudocubic planes as shown in **Fig. R10**, where alternating AO and BO₂ layers are directly visible. We can see that the CTO terminations are indeed same as their substrate terminations, giving an 8 u.c. of CTO films. We also notice that there is a weak electron density distribution above the outer most CTO layers. This suggests that there are incomplete atomic layers arising from the small imperfections during the growth control.

We also performed ABF-STEM on (110) pseudocubic zone axis on the three systems. However, we notice the topmost 1-2 u.c. of CTO films can be easily amorphized by the gold deposition on top of the film during the sample preparation process, which is intended for eliminating the charging effect by the electron beam. Thus, the termination of the CTO films cannot be revealed by STEM, as shown in **Fig. R11**.

These discussions are now captured in the supplementary section.

Figure R10 | (110) pseudocubic planes of COBRA electron densities for **a** CTO/NGO **b** CTO/DSO and **c** CTO/LSO.

Figure R11 | (110) pseudocubic planes of ABF-STEM images for **a** CTO/NGO **b** CTO/DSO and **c** CTO/LSO. The top most 1-2 u.c. of CTO layers are amorphized during the gold deposition process.

Minor Corrections

12. What are the units of L in Figure 1(d)? Are they in orthorhombic NGO units or the 2x2x2 pseudocubic unit cell? If so, that should be defined in the caption and tick marks included on the horizontal axis to identify the peaks.

The reciprocal units in **Fig. 1d** are under 2x2x2 pseudocubic notation, this is specified in the revised manuscript and changes in **Fig. 1d** has been made accordingly to reflect this fact.

13. Could the authors clarify whether the experimental angular values in Supplementary Table 2 are averaged over the entire film?

Yes, the angular values in **Supplementary Table 2** are the averages over the entire film. This is specified in the revised caption for the table.

14. Supplementary note 1 and Supplementary Figure 1 should state the H,K,L values are in terms of the 2x2x2 pseudocubic unit cell.

Thanks for the suggestion. This is specified in the revised **Supplementary Note 1** and caption of **Supplementary Figure 1**.

Reviewer #2 (Remarks to the Author):

In the manuscript NCOMMS-18-16565, the authors study three ultrathin CaTiO₃ films on different substrates and reconstruct 3D oxygen tilts and polar displacements using coherent Bragg rod analysis. Atomic imaging of those thin films using Cs-STEM and DFT results were also performed to give understanding to the observations. The authors observe the highest polar to non-polar transition (900 K) in their ultrathin CaTiO₃/LSAT film as compared to reported thicker one (140-190 K); they reveal an unexpected out-of-plane polar displacement; they also explored the intertwining roles of tilt epitaxy, substrate strain, and substrate surface terminations. The experimental aspects of this work are outstanding and essential for moving forward our general understanding of tilt epitaxy engineering in ultrathin perovskite films. Overall, I think this is a very novel finding that will be of broad interest.

(1) In studies such as this, it is nice to know the presence of switchable P-E loops in those extraordinary ultrathin CaTiO₃ films. In particular, the authors term ferroelectric transition in the manuscript. If the polarity cannot be switched, the system is just polar. I think that the switching characteristics have to be provided if the authors claim a ferroelectric phase and possible applications.

We agree with the reviewer that it will be great to demonstrate the switchable polarization in these ultrathin CTO films (~3 nm). However, the dielectric measurements on films with such thin thickness are technically very challenging. Our previous attempts on such thin films were not successful. Instead, we observed clear P-E loop along in-plane direction on much thicker CTO films (~20 nm) on NGO, DSO, and LSAT substrates at ~20 K, as shown in **Fig. R12**. These P-E loops demonstrate the thick CTO films on above substrates are indeed ferroelectric with in-plane remnant polarizations of 1~5 μC/cm². However, the observed polarization signal in **Fig. R12** should be dominated by thick CTO film far away from the film-substrate interface. Hence, there is still no direct evidence to support the polarizations in ultrathin CTO films are switchable.

To reflect this fact, we change the words 'ferroelectric transition' to 'polar transition' in the main text for better scientific accuracy. Despite of lacking evidence for ferroelectricity in these ultrathin films, we believe that the revealing of intertwining roles of tilt epitaxy, substrate strain, and substrate surface

terminations, as well as the method discussed in this work should be generally applicable to epitaxial heterostructures, and not limited to ferroelectrics.

Figure R12 | P-E loop measurements on thick CTO films (~20 nm) on NGO, DSO, and LSAT substrates at ~20 K. Clear ferroelectric behavior is observed in all three systems.

(2) The authors reveal the unexpected out-of-plane polar displacement in ultrathin CaTiO₃ films and consider it as interfacial valence mismatch effect. On the other hand, the depolarization field may suppress the out-of-plane ferroelectricity in their ultrathin films as well. The authors need to provide more detailed discussion on this issue.

We first would like to point out that the intriguing out-of-plane polarization in these ultrathin CTO films originates from both interfacial valence mismatch and tilt epitaxy effects. This tilt epitaxy effect is detailed in the response to the 1st question of reviewer 1 (as a *domino* effect of both tilt distortion and electrostatics starting from near the interface, but also extending throughout the film thickness), which is the underlying driving force responsible for the non-zero out-of-plane polarization in these ultrathin CTO films. The valence mismatch effect is more prominent near the film-substrate interface, enhancing the out-of-plane polarization in the very first ~2 u.c., as reviewed by both COBRA and DFT results in main **Figs. 3a-c**. (COBRA result on CTO/LSAT is not as clear.) This valence mismatch enhanced polarization is very localized near the film-substrate interface. Apart from the topmost CTO u.c., where surface effects may present, on approaching the film surface, the out-of-plane polarizations shows gradual decrease in magnitude. This behavior can be understood to arise from the depolarization field and the relaxation of the tilt epitaxy effects away from the interface.

This discussion is briefly captured in the main text.

(3) In epitaxial heterojunctions, the film-substrate misfit strains often can be relaxed by the crystallographic defects such as dislocations with typically few hundred nanometers or be accommodated by uniform deformations of unit cells. With a relatively broad probe area of X-ray (compared to Cs-STEM), those imperfections may significantly affect the Bragg peaks and not been considered in the phase retrieval algorithm fitness. As stated by the authors, they deal with the most complex low symmetry and bulky structural reconstruction to date by the technique. The authors may briefly comment on this concern.

This question partially relates to the strain states of the CTO films, which is discussed in the response to the 10th question of reviewer 1. The main conclusion from our X-ray results is that all the CTO films are under fully commensurate strained states. As the reviewer pointed out, the misfit strain is often relaxed by defects, which usually form at a critical thickness where the energy penalty by defect formation is smaller than the elastic energy of straining the film. Our previous experience suggests the critical thickness is typically >10 nm for these CTO film, which can be much larger under smaller strain state. The ultrathin (~3 nm) film thickness used in this work is well below the critical thickness, and fully strained state is expected.

We would also point out that the phase-retrieval algorithm used here are based on kinematic diffraction approximation, where strong X-ray diffractions at Bragg peaks of substrates cannot be dealt with. However, the CTO film Bragg peaks are much weaker and located slightly away from the substrate peaks, especially for the case of CTO/DSO with 3.3% tensile strain. Thus, kinematic diffraction is still applicable to these thin films even at their Bragg peak positions. In addition, the severe defect formation will strongly alter the in-plane periodicity of the films, leading to a blurred electron density due to the in-plane folding effect, which is not observed in our results shown in main **Fig. 2**. Finally, our STEM work does not reveal any dislocations in the many areas that were imaged. (with the caveat that STEM is a local probe and cannot cover the entire film)

This issue has now been briefly discussed in the main text.

(4) There is no details of “Methods” and DFT calculations in the main text and supplementary information.

The method section of previous manuscript was in a separate file. Following the reviewer’s suggestion, it has been moved to the end of current main manuscript.

(5) The authors may recheck references more carefully. For example pages are missing in ref 41 and 43. On the other hand, the authors cited a few theoretical works about the octahedral rotation-induced ferroelectricity while experimental demonstrations of switchable improper mode involving octahedral rotations are well recognized recently, e.g. Bousquet et al., Nature 452, 732 (2008) and Oh et al., Nature Materials 14, 407 (2015).

Thanks for these corrections. Those corrections have been made in revised manuscript.

Reviewer #3 (Remarks to the Author):

June 22, 2018

Brief summary of the manuscript:

The authors present a detailed atomic structural study investigating substrate-induced octahedral tilt patterns in ultra-thin films of CaTiO₃ on three different substrates with differing symmetries and lattice mismatches. The main results are drawn from a comprehensive 3-dimensional analysis of the electron density reconstruction over the whole film thickness, including the top layers of the substrate material, using the iterative phase retrieval method COBRA. The established COBRA method has been adapted by the authors to work also on systems with a low in-plane symmetry, and the structural results obtained from the 3D electron density maps are corroborated by complementary experimental techniques, particularly STEM and optical SHG, as well as by comparison to density functional theory.

General remarks:

The novelty of the described research is mostly technical in nature: The reported results certainly represent some of the most complete atomic structural determinations of thin film systems, particularly

when taking the low symmetry of these systems into account. The structural insights gained from these measurements can help to disentangle the individual contributions of surface termination and symmetry, lattice strain, and structural distortions across the interface caused by a continuous change in the oxygen octahedral network orientation and tilt patterns. Thus, the method provides crucial experimental input for a basic understanding of the response of these thin film systems subjected to different substrate constraints, which should ultimately allow for a tailored design of specific desired film properties.

The observed effects in this study are not surprising or all new by themselves, and the concept of induced or coherent structural distortions across interfaces in thin film or multilayer systems has been around for decades. I would thus encourage the authors to devote an additional paragraph in the introductory part of the manuscript to specifically review the different cases that have been discussed in the literature thus far, in particular with regards to the octahedral tilt patterns. A number of the relevant papers are cited in the introduction in "bulk form" (17-26), but their relation and relevance with respect to the present study is not discussed.

We agree with the reviewer that a paragraph briefly summarizing previous literature will help readers better understand the current status of 'tilt epitaxy' and the impact of our work. Hence, the following discussion on this issue is added to the first introductory paragraph of the paper:

'... Control of octahedra tilts in complex oxides via film-substrate interface design, or "tilt epitaxy", has been predicted to be a powerful knob for tuning various functional properties, including inversion symmetry breaking [26-28], magnetism [18,22,29], and electronic orders [30]. Although the tilt epitaxy promises a potential wonderful route of designing these functionalities in various materials, the experimental reports on realizing tilt epitaxy are still very limited. So far, the tilt epitaxy has been used to stabilize polar distortions into metallic ultrathin nickelates films [20] and to manipulate magnetic anisotropy in SrRuO₃ [18,29] and La_{2/3}Sr_{1/3}MnO₃ [22]. However, in these works, only in-phase octahedra tilt along one of the three crystallographic axes are experimentally resolvable. Moreover, strain and substrate termination effects can convolute with tilt epitaxy, which remain unexplored. In general, there are three outstanding challenges in implementing tilt epitaxy...'

From a technical point of view, the manuscript is very solid and well written, although it should be revised for minor grammatical and language issues. Figures are clear and illustrate well the contents described in the text. The relevant literature is adequately cited. Conclusions and interpretations of the structural results are well supported by experimental results (note that I am not an expert in DFT and SHG measurements, so I am unable to assess those corresponding results and interpretations in detail).

My main concern with the manuscript is related to the novelty, significance, and impact of the reported findings in the scientific community and in relation to the publication requirements by Nature Communications. As detailed above, the technical achievement in obtaining 3D ED maps at this level of detail for low-symmetry systems is undisputed. However, the method is in principle based on a technique (COBRA) which has been around for 15 years (a good number of citations to earlier works are cited in the manuscript), and which has been further refined in this work (generalization to lower symmetry systems) to obtain the presented results. Although a number of excellent structural studies have been published with COBRA since its inception, its broader impact has been limited so far. To be completely honest, I am not fully convinced that the present manuscript will change this situation significantly, although the authors state in their conclusions that "With the development of high-energy surface X-ray diffraction that enables the rapid capture of large portions of 3D Bragg rods in reciprocal space, COBRA measurements will become much more efficient and widespread in their application to oxide thin films and heterostructures". I would also argue that the last part of the last sentence, "This work will motivate progress ..., and more broadly, provide a powerful non-destructive tool with atomic resolution for probing the electron density of complex functional interfaces.", is somewhat exaggerated in this context. In my view, this work rather provides an important and very beneficial improvement to a tool which has, however, been available since long before the present work. I would therefore ask the authors to clarify

these statements and substantiate their claim why the technique would suddenly gain so much in importance through the work presented in this manuscript. Note again that I am not disputing the validity of the presented results, I am merely questioning the impact of this particular work with regards to making the experimental technique itself more accessible to a broader community working on complex functional interfaces and beyond.

We thank the reviewer's recognition of our work being 'technically solid and well written' and 'the achievement in obtaining 3D ED maps at this level of detail for low-symmetry systems is undisputed'. The other reviewers appear to agree: 'These results will be of general interest to the field of oxide interfaces.' and that 'The experimental aspects of this work are outstanding and essential for moving forward our general understanding of tilt epitaxy engineering in ultrathin perovskite films. ... this is a very novel finding that will be of broad interest.'

The critical comments to improve the manuscript are greatly appreciated! We would like to further elaborate on the possible broad impact of our work in the following two aspects (without hopefully overdoing it): materials science and experimental technique.

Impact in materials science: As discussed in the manuscript, the engineering of various materials functionalities using tilt epitaxy has been predicted to be generally effective and promising in complex oxides. However, the intertwining roles of tilt epitaxy, strain, and substrate termination can significantly complicate the study. Existing experimental reports on the tilt epitaxy effect are very limited, and are solely supported by STEM results on one in-phase tilt axis out of the three tilt axes. This is due to the intrinsic limitation of 2D projection provided by STEM. When taking strain and substrate termination into account, although important, there has been no study discussing all these effects in a single material system.

For the first time, we have tried to establish a comprehensive understanding of these three important effects (strain, substrate termination and tilt epitaxy) in ultrathin CaTiO_3 films using COBRA method. The clear 3D electron densities greatly benefit the interpretation of the roles of each in realizing emergent properties. We reveal that the tilt epitaxy effect stabilizes the polar phase up to a much higher transition temperature, modulates the in-plane polarization together with epitaxial strain effect, and leads to an out-of-plane polarization with direction dictated by the substrate termination and valence mismatch.

Impact in experimental technique: We agree with the reviewer that the COBRA method has been around for 15 years, but its experimental application is still limited to certain systems. This is mainly due to two reasons.

First, COBRA experiments require advanced high brilliance synchrotron X-ray source to achieve high signal-to-noise ratio for measuring weak diffraction intensities distributed between Bragg peaks. Usually, a large number of crystal truncation rods need to be measured to achieve good spatial resolution in three dimensions, which is often very time consuming, especially for the low symmetry systems we are dealing with in this work. The above experimental difficulties will be much improved with the advances in synchrotron techniques. Particularly, the development of high energy X-ray surface diffraction capability and large size area detector in the third-generation and future synchrotron sources will enable rapid capturing of crystal truncation rods in large reciprocal volumes, thus greatly facilitating the COBRA data acquisition. A recent test at a high energy X-ray beamline at the Advanced Photon Source has demonstrated that the data collection efficiency can be improved by more than one order of magnitude even for a low-symmetry oxide heterostructure like in our current study. Such improvement will allow much more accessibility of the techniques for many scientific communities beyond complex oxides.

Secondly, there are limited groups in the world working on developing the COBRA method although its utilizations have been demonstrated by many leading groups in the field of oxide heterostructures. In recent years, a few other groups studying non-oxide reduced-dimensional systems such as transitional

metal dichalcogenide 2D materials have also applied the COBRA method to determine the interfacial atomic structures. The development and standardization of COBRA routines still need years of dedication of highly specialized experts from materials science, X-ray communities and computer science. One of our important future directions is to develop an automated COBRA data collection and analysis platform with graphical user interfaces to facilitate users from various backgrounds to be able to use the method. In the meantime, we hope that the generalization of COBRA routine to low symmetry systems and its unique application to the tilt epitaxy effect demonstrated by this work would attract more attention from the scientific community to the COBRA method and its capability. We hope the reviewer would agree with us that this work is still technically important, and we certainly hope that it will be an important stepping stone in the development of the COBRA method. It is a vision, and we are trying our best to make that vision come true.

In light of the above concerns, I cannot recommend the manuscript as is for publication in Nature Communications for the sole reason that I don't think the manuscript fulfills the requirements for novelty, significance for the field and interest of a broad audience. Since the manuscript is of very high quality in all other aspects, I recommend that the authors either attempt to better justify the expected broad impact of their paper, or submit to a more specialized journal in the field of complex functional oxide systems or thin film structure determinations.

We hope the reviewer may kindly reconsider this opinion, given 5 years of dedication to make this work possible, the quality of the work, and a vision ahead of making this powerful technique accessible for the users. This work is a stepping stone that highlights what is possible with this technique, and continued efforts on our part and that of other groups will make it happen sooner than later.

Miscellaneous issues and comments:

1) Figure 2: This is a beautiful figure illustrating the 3D data obtained from the COBRA method. However, a few of the important details are hard to capture simply due to the fact that the corresponding graphics are very small. For example, the pie charts indicating the octahedral rotation angles next to the 3D representation of the ED map are very hard to discern, particularly on a printed piece of paper. The same is true for the insets highlighting, for example, the dashed elliptical lines outlining the oxygen atom positions. These lines are barely visible, and their dashed nature is impossible to see. I would suggest to enlarge these important details if possible.

We thank the reviewer for these important suggestions. The pie charts along the sides of the electron densities in **Figs. 2a-c** have been enlarged to their maximum available space without overlapping with other objects in the revised version of the figure. We will also use the full width of a page for this figure to make sure the details are displayed with proper sizes in the printed version. The dashed elliptical lines for the oxygen atoms in **Fig. 2e** have been replaced by solid elliptical lines in the revised manuscript. With above efforts, these details in the revised **Fig. 2** should be much easier to visualize.

REVIEWERS' COMMENTS:

Reviewer #1 (Remarks to the Author):

My concerns regarding the estimation of errors and the presentation of additional data to confirm the octahedral rotational patterns have been adequately addressed. While, the application of the phase retrieval technique to a low dimensional system is new in a sense, the constraints that the substrate has to be preferably single domain may limit its general application to other systems where twinning frequently occurs. Additionally, since one has to identify the symmetry of the film by the measurements of other peaks (which is the current state of the art), in the abstract,

"..These maps ... well as three independent oxygen octahedral tilts (a-b+c-) in ultrathin films on substrates that also possess such tilts, the first such feat.... "

should be qualified with a statement indicating the COBRA technique is complemented with TEM and the measurement of film superstructure reflections.

I recommend the current version of the manuscript for publication since I believe the results do contribute to our understanding of the coupling of interfaces.

Reviewer #2 (Remarks to the Author):

I have checked the author's reply and the revised manuscript. The authors have addressed my questions reasonably well, so I suggest you to accept it as it is.

Reviewer #3 (Remarks to the Author):

The authors have very carefully and comprehensively responded to the comments from all referees, providing additional analysis and interpretations. This goes to show again that the work is of high technical quality.

My main concern was with the novelty of the reported technical developments and their broader impact on the community. I do agree with the authors that the scientific impact of this work in the field of materials science in general and in thin film property engineering in particular is significant. This view is also clearly shared by the other reviewers.

However, I remain skeptical about the impact of the technique itself. The authors themselves admit in their rebuttal that "we hope that the generalization of COBRA routine ... demonstrated by this work would attract more attention from the scientific community", and that "we certainly hope that it will be an important stepping stone in the development of the COBRA method". Altogether, this sounds more like an incremental step ("stepping stone") rather than a technical breakthrough. And it was this lack of direct technical impact that I was criticizing. In the long run, only time will tell whether the presented improvements to the technique will meet the high expectations of the authors in terms of its widespread impact.

Nevertheless, upon careful consideration, my first review was probably too focused on this technical impact, somewhat neglecting the scientific impact. I am therefore happy to revise my original assessment, and recommend the manuscript for publication. The revisions provided by the authors have further strengthened the scientific case.

We thank the reviewers for their valuable suggestions and recommendations. We have made corresponding changes in the revised manuscripts.

Reviewer #1 (Remarks to the Author):

My concerns regarding the estimation of errors and the presentation of additional data to confirm the octahedral rotational patterns have been adequately addressed. While, the application of the phase retrieval technique to a low dimensional system is new in a sense, the constraints that the substrate has to be preferably single domain may limit its general application to other systems where twinning frequently occurs. Additionally, since one has to identify the symmetry of the film by the measurements of other peaks (which is the current state of the art), in the abstract,

"..These maps ... well as three independent oxygen octahedral tilts (a-b+c-) in ultrathin films on substrates that also possess such tilts, the first such feat.... "

should be qualified with a statement indicating the COBRA technique is complemented with TEM and the measurement of film superstructure reflections.

I recommend the current version of the manuscript for publication since I believe the results do contribute to our understanding of the coupling of interfaces.

We thank the reviewer for all the valuable suggestions and the recognitions of our work. We agree that measuring the film superstructure reflections is very helpful in identifying the symmetry of the epitaxial films and the tilt patterns. Following the reviewer's suggestion, we made changes in the revised abstract as below:

"...The results are complemented with aberration-corrected transmission electron microscopy, film superstructure reflections, and are in excellent agreement with density functional theory...."

Reviewer #2 (Remarks to the Author):

I have checked the author's reply and the revised manuscript. The authors have addressed my questions reasonably well, so I suggest you to accept it as it is.

We thank the reviewer for recommending our work for publication.

Reviewer #3 (Remarks to the Author):

The authors have very carefully and comprehensively responded to the comments from all referees, providing additional analysis and interpretations. This goes to show again that the

work is of high technical quality.

My main concern was with the novelty of the reported technical developments and their broader impact on the community. I do agree with the authors that the scientific impact of this work in the field of materials science in general and in thin film property engineering in particular is significant. This view is also clearly shared by the other reviewers.

However, I remain skeptical about the impact of the technique itself. The authors themselves admit in their rebuttal that "we hope that the generalization of COBRA routine ... demonstrated by this work would attract more attention from the scientific community", and that "we certainly hope that it will be an important stepping stone in the development of the COBRA method".

Altogether, this sounds more like an incremental step ("stepping stone") rather than a technical breakthrough. And it was this lack of direct technical impact that I was criticizing. In the long run, only time will tell whether the presented improvements to the technique will meet the high expectations of the authors in terms of its widespread impact.

Nevertheless, upon careful consideration, my first review was probably too focused on this technical impact, somewhat neglecting the scientific impact. I am therefore happy to revise my original assessment, and recommend the manuscript for publication. The revisions provided by the authors have further strengthened the scientific case.

We are delighted to learn that the reviewer recommends our work for publication. We also greatly appreciate the reviewer's critical comments on the technical impact of this work.